# Covariances for Free: Exploiting Mean Distributions for Training-free Federated Learning

**Dipam Goswami**[1,2]   **Simone Magistri**[3]   **Kai Wang**[2,5,6*]
**Bartłomiej Twardowski**[1,2,4]   **Andrew D. Bagdanov**[3]   **Joost van de Weijer**[1,2]

[1]Department of Computer Science, Universitat Autònoma de Barcelona, Spain
[2]Computer Vision Center, Barcelona, Spain
[3]Media Integration and Communication Center (MICC), University of Florence, Italy
[4]IDEAS Research Institute, Warsaw, Poland    [5]City University of Hong Kong
[6]Program of Computer Science, City University of Hong Kong (Dongguan)
`dgoswami@cvc.uab.es, simone.magistri@unifi.it, kai.wang@cityu-dg.edu.cn`

## Abstract

Using pre-trained models has been found to reduce the effect of data heterogeneity and speed up federated learning algorithms. Recent works have explored training-free methods using first- and second-order statistics to aggregate local client data distributions at the server and achieve high performance without any training. In this work, we propose a training-free method based on an unbiased estimator of class covariance matrices which only uses first-order statistics in the form of class means communicated by clients to the server. We show how these estimated class covariances can be used to initialize the global classifier, thus exploiting the covariances without actually sharing them. We also show that using only within-class covariances results in a better classifier initialization. Our approach improves performance in the range of 4-26% with exactly the same communication cost when compared to methods sharing only class means and achieves performance competitive or superior to methods sharing second-order statistics with dramatically less communication overhead. The proposed method is much more communication-efficient than federated prompt-tuning methods and still outperforms them. Finally, using our method to initialize classifiers and then performing federated fine-tuning or linear probing again yields better performance. Code is available at `https://github.com/dipamgoswami/FedCOF`.

## 1   Introduction

Federated learning (FL) is a widely used paradigm for distributed learning across multiple clients. In FL, each client trains their local model on their private data and then sends model updates to a common global server that aggregates this information into a global model. The objective is to learn a global model that performs similarly to a model jointly trained on all the client data. A major concern in existing FL algorithms [37] is the poor performance when the client data is not identically and independently distributed (iid) or when classes are imbalanced between clients [56, 32, 1, 21]. In [35], the authors showed that client drift in FL is mainly due to drift in client classifiers which optimize to local data distributions, resulting in forgetting knowledge from clients of previous rounds [30, 5]. Another challenge in FL is the partial participation of clients in successive rounds [32], which becomes particularly acute with large numbers of clients [43, 20]. To address these challenges, recent works have focused on algorithms to better tackle data heterogeneity across clients [35, 47, 29, 11].

---

[*]Corresponding author.

39th Conference on Neural Information Processing Systems (NeurIPS 2025).

Table 1: FedNCM [29] shares only class means $\hat{\mu}_{k,c}$ and has minimal communication. Fed3R [11] requires sum of class features $B_k$ and feature matrix $G_k$ from all clients, thereby increasing the communication cost by $d^2K$. We propose FedCOF, which shares only class means and estimates a global class covariance $\hat{\Sigma}_c$ to initialize the classifier weights. Note that only a small subset of all classes are present in each client. For simplicity, we show the upper bound of communication cost here where $C'$ denotes the maximum number of classes present in a single client.

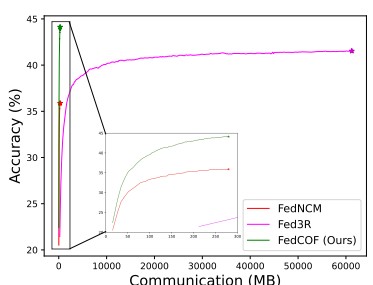

Figure 1: Accuracy vs. communication cost using pre-trained MobileNetv2 on iNaturalist-Users-120K. FedCOF outperforms Fed3R while having the same communication cost as FedNCM.

| Method | Client Shares | Server Uses | Comm. Cost |
|--------|---------------|-------------|------------|
| FedNCM | $\{\hat{\mu}_{k,c}, n_{k,c}\}_{c=1}^{C'}$ | $\{\{\hat{\mu}_{k,c}, n_{k,c}\}_{c=1}^{C'}\}_{k=1}^{K}$ | $dC'K$ |
| Fed3R | $G_k, B_k$ | $\{G_k, B_k\}_{k=1}^{K}$ | $(dC' + d^2)K$ |
| FedCOF | $\{\hat{\mu}_{k,c}, n_{k,c}\}_{c=1}^{C'}$ | $\{\{\hat{\mu}_{k,c}, n_{k,c}\}_{c=1}^{C'}\}_{k=1}^{K}, \{\hat{\Sigma}_c\}_{c=1}^{C'}$ | $dC'K$ |

Motivated by results from transfer learning [15], several recent works on FL have studied the impact of using pre-trained models and observe that it can significantly reduce the impact of data heterogeneity [29, 38, 47, 6, 41, 45, 35, 46, 54]. An important finding in several of these works is that sending local class means to the server instead of raw features is more efficient in terms of communication costs, eliminates privacy concerns, and is robust to gradient-based attacks [6, 57]. The authors of [47] used pre-trained models to compute and then share class means as the representative of each class, and in [29] the authors showed that aggregating local means into global means and setting them as classifier weights (FedNCM) achieves very good performance without any training. FedNCM incurs very little communication cost and enables stable initialization. Recently, the authors of Fed3R [11] explored the impact of sharing second-order feature statistics from clients to server to solve the ridge regression problem [4] in FL and improve over FedNCM. The sharing of second-order statistics has also previously been explored for classifier calibration after federated optimization [35].

Although it is evident that exploiting second-order feature statistics results in better and more stable classifiers, it poses new problems. Notably, sharing second-order statistics for high-dimensional features from clients to the server significantly increases communication overhead and exposes clients to privacy risks [35, 11]. To reap the benefits of second-order client statistics, while at the same time mitigating these risks, in this paper we propose Federated learning with COvariances for Free (FedCOF) which only communicates class means from clients to the server (see Table 1). We show that, from just these class means and the mathematical relationship between their covariance and the class covariance matrices, we can compute a provably unbiased estimator of global class covariances on the server. FedCOF is a training-free method that exploits pre-trained feature extractors. It uses the same communication budget as FedNCM while delivering performance comparable to or even superior to Fed3R (see Figure 1). Additionally, we show how to improve classifier initialization using only within-class covariances, setting the classifier weights based on aggregated class means and our estimated class covariances. To summarize, our main contributions are:

- We propose FedCOF, a training-free method that exploits a *provably unbiased estimator of class covariances*, which requires only class means from clients, thus avoiding the need to share second-order statistics, reducing communication costs and mitigating privacy concerns.

- We show how FedCOF leverages *within-class covariances*, estimated solely from client means, to initialize the global classifier, yielding more stable and better-conditioned solutions than using both within- and between-class scatter matrices.

- We validate FedCOF across multiple FL benchmarks, including the non-iid iNaturalist-Users-120K dataset, achieving state-of-the-art results with lower communication cost than methods using second-order statistics, and showing superior performance to recent federated prompt-tuning approaches while also serving as an effective initialization for subsequent federated optimization methods such as fine-tuning and linear probing.

## 2   Related Work

**Federated learning.** FL focuses on neural network training in distributed environments [55, 53]. Initial works like FedAvg [37] proposed training by averaging of distributed models. Later works

focus more on non-iid settings, where data among the clients is more heterogeneous [32, 20, 51, 33]. FedNova [52] normalizes local updates before averaging to address objective inconsistency. Scaffold [22] employs control variates to correct drift in local updates. FedProx [31] introduces a proximal term in local objectives to stabilize the learning process. [42] proposed use of adaptive optimization methods, such as Adagrad, Adam and Yogi, at the server side. While CCVR [35] proposed a classifier calibration method that aggregates class means and covariances from clients, other recent works [34, 8, 39, 23] proposed using fixed classifiers inspired by the neural collapse phenomenon. After federated training with fixed classifiers, FedBABU [39] proposed to fine-tune the classifiers and SphereFed [8] proposed a closed-form classifier calibration.

**FL with pre-trained models.** While conventional FL methods start training from scratch without any pre-training, we focus on the FL setting using pre-trained models. FedFN [24] recently highlighted that using pre-trained weights can sometimes negatively impact performance. However, there has been increasing interest in incorporating pre-trained, foundation models into federated learning. Multiple works propose using pre-trained weights which reduces the impact of client data heterogeneity and achieves faster model convergence [38, 47, 6, 41, 45, 54]. Federated prompt-tuning methods [54] proposed to reduce the communication cost by tuning only prompts in a federated manner using pre-trained ViT models. Recently, it has been shown that *training-free methods* using pre-trained networks, achieve strong performance without any training by exploiting feature class means [29] or second-order feature statistics [11]. Another recent work [2] proposed sharing Gaussian mixture models (multiple sets of means and covariances) for each class from clients to server for one-shot FL. In this work, we propose a training-free method with pre-trained models that estimates class covariances from only client means for initializing the global classifier.

## 3 Preliminaries

In the FL setting we assume $K$ clients, each with a local dataset $D_k = (X_k, Y_k)$, for $k \in \{1, ..., K\}$, and denote by $N$ the total number of images across all clients. The model is composed of a feature extractor $f$, parameterized by $\theta$, which maps images to $d$-dimensional embeddings, and a classifier $h : \mathbb{R}^d \to \mathbb{R}^C$, parameterized by $W$, where $C$ is the total number of classes. In federated optimization, each client trains its local model on its private data $D_k$ and transmits only intermediate information – such as parameter updates or feature statistics – while a central server aggregates these signals to minimize a global objective, without revealing raw data [25].

With the growing availability of high-quality pre-trained models, recent works have focused on scenarios in which all clients are initialized with the same pre-trained feature extractor [6, 29, 38, 47, 11, 54]. Notably, *training-free* methods [11, 29] – in which only the global classifier is initialized using client feature statistics, without training the feature extractor – achieve strong performance. They often outperform federated full fine-tuning methods, such as FedAdam and FedAvg, which train client backbones and classifiers, starting from the same shared pre-trained feature extractor but with randomly initialized classifiers, at a fraction of the communication and computation costs. Moreover, this training-free initialization can serve as an effective starting point, improving the performance of federated full fine-tuning. We now discuss recent training-free methods.

**Federated NCM.** FedNCM [29] employs a Nearest Class Mean (NCM) classifier, where the global linear classifier weights for class $c$ (denoted as $W_c$) are initialized as $\hat{\mu}_c / \|\hat{\mu}_c\|$. Here, $\hat{\mu}_c$ represents the global class mean aggregated from the local client class means $\hat{\mu}_{k,c}$ as follows:

$$\hat{\mu}_c = \frac{1}{N_c} \sum_{k=1}^{K} n_{k,c} \, \hat{\mu}_{k,c}; \quad \hat{\mu}_{k,c} = \frac{1}{n_{k,c}} \sum_{x \in X_{k,c}} f(x), \tag{1}$$

where $X_{k,c}$ is a subset of $X_k$ having images of class $c$, $n_{k,c}$ refers to number of images in $X_{k,c}$, and $N_c = \sum_{k=1}^{K} n_{k,c}$ is the number of images of class $c$ across all clients.

**Federated Ridge Regression.** While FedNCM exploits only class means, Fed3R [11] recently proposed using ridge regression which requires second-order feature statistics from all clients to initialize the global classifier, leading to improved performance compared to FedNCM. The ridge regression problem aims to find the optimal weights that minimize the following objective:

$$W^* = \arg \min_{W \in \mathbb{R}^{d \times C}} \|Y - F^\top W\|^2 + \lambda \|W\|^2, \tag{2}$$

where $F \in \mathbb{R}^{d \times N}$ is the feature matrix extracted using a pre-trained model and $Y \in \mathbb{R}^{N \times C}$ contains one-hot encoded labels for the $N$ features over $C$ classes. The closed-form solution is given by:

$$W^* = (G + \lambda I_d)^{-1} B, \tag{3}$$

where $G = FF^\top \in \mathbb{R}^{d \times d}$, $B = FY \in \mathbb{R}^{d \times C}$, $I_d \in \mathbb{R}^{d \times d}$ is the identity matrix, and $\lambda \in \mathbb{R}$ is a hyperparameter. In Fed3R, each client $k$ computes two local matrices $G_k = F_k F_k^\top \in \mathbb{R}^{d \times d}$ and $B_k = F_k Y_k \in \mathbb{R}^{d \times C}$, where $F_k$ and $Y_k$ are the feature matrix and the labels of client $k$, and then sends them to the global server. The server aggregates these matrices as $G = \sum_{k=1}^{K} G_k$, $\quad B = \sum_{k=1}^{K} B_k$ to compute $W^*$, which is normalized and used to initialize the global classifier.

# 4 Federated Learning with COvariances for Free (FedCOF)

In this section, we derive our FedCOF method which is based on a provably unbiased estimator of class covariances and requires transfer of only class means from clients to server.

## 4.1 Motivation

**Communication cost.** While Fed3R is more effective than FedNCM, it requires each client to send an additional $d \times d$ matrix, significantly increasing the communication overhead by $d^2 K$ compared to FedNCM which only shares the class means (see Table 1). Fed3R scales linearly with number of clients and quadratically with the feature dimension. FedPFT [2] shares multiple sets of $(\mu_{k,c}, \Sigma_{k,c})$ from clients, further increasing communication costs. Considering cross-device FL settings [20], having millions of clients, the communication cost required for these methods would be enormous.

Recent works [26, 54] focus on parameter-efficient federated fine-tuning where the per-round communication costs are significantly reduced, enabling applications in low-bandwidth communication settings. For example, recent work [26] shows that using pre-trained language models (RoBERTa-base) can yield performance similar to full fine-tuning while using rank-1 LoRA updates and thereby reducing communication cost by 99.8%. Sharing a similar goal of reducing communication costs, we propose an unbiased covariance estimator without sharing client covariances for training-free FL.

**Potential privacy concerns.** Sharing only class means provides a higher level of data privacy compared to sharing raw data, as prototypes represent the mean of feature representations. It is not easy to reconstruct exact images from prototypes with feature inversion attacks [35]. As a result, sharing class means is common in many recent works [47, 46, 45, 29]. On the other hand, Fed3R shows that sharing second-order statistics improves performance compared to sharing class means. However, this could expose the feature distribution of clients to the server since all clients employ the same frozen pre-trained model to extract features [11]. Sharing covariances makes clients more vulnerable to attacks if secure aggregation protocols are not implemented [3].

## 4.2 Estimating Covariances Using Only Client Means

Our method leverages the statistical properties of sample means to derive an unbiased estimator of the class population covariance based only on per-client class means (see Figure 2). We model the global features of each class $c$ as drawn from a multivariate distribution with mean $\mu_c$ and covariance $\Sigma_c$, and assume that the local features for class $c$ computed by each client using the shared *frozen* pre-trained model are iid. Under this assumption, class features in a client form a random sample from the class population.

For client $k$, let $\{F_{k,c}^j\}_{j=1}^{n_{k,c}}$ denote the feature vectors of class $c$, where $n_{k,c}$ is the number of samples from class $c$ assigned to client $k$. The sample mean of these features, $\overline{F}_{k,c} = \frac{1}{n_{k,c}} \sum_{j=1}^{n_{k,c}} F_{k,c}^j$, is itself a random variable with expectation and variance given by:

$$\mathbb{E}[\overline{F}_{k,c}] = \mu_c, \quad \mathrm{Var}[\overline{F}_{k,c}] = \frac{\Sigma_c}{n_{k,c}} \tag{4}$$

This well-known result – whose proof we include in Appendix B for reference – implies that the variance of sample means over subsets of size $n_{k,c}$ reflects the underlying class covariance. In principle, by assigning multiple subsets of $n_{k,c}$ features to a single client and computing the empirical covariance of their means, one could recover $\Sigma_c$, since $\Sigma_c = n_{k,c} \mathrm{Var}[\overline{F}_{k,c}]$.

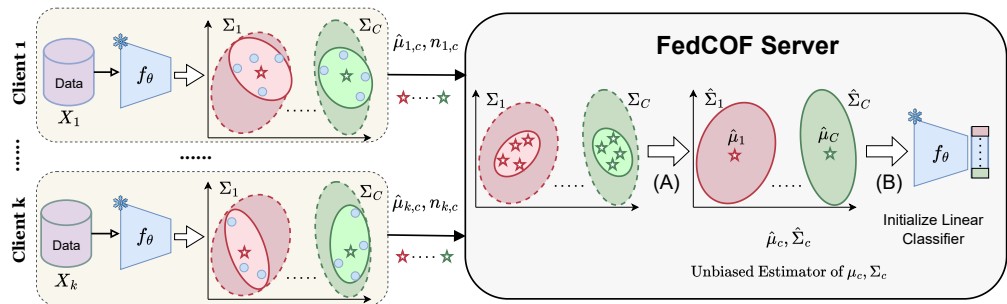

Figure 2: **Federated Learning with COvariances for Free (FedCOF)**. Each client $k$ communicates only its class means $\hat{\mu}_{k,c}$ and counts $n_{k,c}$. On the server side, (A) we use a provably unbiased estimator $\hat{\Sigma}_c$ (denoted by solid lines) of population covariance $\Sigma_c$ (denoted by dashed lines) based on the received class means (see Section 4.2). (B) We initialize the linear classifier using the estimated second-order statistics and remove the between-class scatter matrix as discussed in Section 4.3.

However, in federated learning data are assigned only once to each client, and there are $K$ clients in the federation, each with $n_{k,c}$ features and $n_{i,c} \neq n_{j,c}$ for $i \neq j$. To estimate the population covariance $\Sigma_c$, we need an estimator that accounts for the contributions of all $K$ clients. In the following proposition, we propose just such an estimator.

**Proposition 1.** *Let $K$ be the number of clients, each with $n_{k,c}$ features, and let $C$ be the total number of classes. Let $\hat{\mu}_c = \frac{1}{N_c} \sum_{j=1}^{N_c} F^j$ be the unbiased estimator of the population mean $\mu_c$ and $N_c = \sum_{k=1}^{K} n_{k,c}$ be the total number of features for a single class. Assuming the features for class $c$ are iid across clients at initialization, the estimator*

$$\hat{\Sigma}_c = \frac{1}{K-1} \sum_{k=1}^{K} n_{k,c} (\overline{F}_{k,c} - \hat{\mu}_c)(\overline{F}_{k,c} - \hat{\mu}_c)^\top \tag{5}$$

*is an unbiased estimator of the population covariance $\Sigma_c$, for all $c \in 1, \ldots, C$.*

*Proof.* To prove that $\hat{\Sigma}_c$ is an unbiased estimator of the population covariance, we show that $\mathbb{E}[\hat{\Sigma}_c] = \Sigma_c$. Under the iid assumption of client feature distribution with a frozen pre-trained model, the class features of each client can be considered as a random sample of size $n_{k,c}$, and the global class features as a sample of size $N_c$. By exploiting the result in Equation (4), each client class mean has $\mathbb{E}[\overline{F}_{k,c}] = \mu_c$ and $\text{Var}[\overline{F}_{k,c}] = \frac{\Sigma_c}{n_{k,c}}$, while the global class mean $\hat{\mu}_c$ has $\mathbb{E}[\hat{\mu}_c] = \mu_c$ and $\text{Var}[\hat{\mu}_c] = \frac{\Sigma_c}{N_c}$. Using this fact and applying the properties of expectation to $\mathbb{E}[\hat{\Sigma}_c]$, we complete the proof. In Appendix C we provide the detailed proof.

**Covariance shrinkage.** Shrinkage[49, 12] stabilizes covariance estimation by adding a scaled identity matrix to the covariance matrices, which regularizes small eigenvalues and reduces estimator variance. This is particularly helpful when the number of samples is smaller than the feature dimensionality, leading to low-rank covariances, and has been recently adopted in continual learning methods leveraging feature covariances [14, 36]. In the FL setup, the covariance estimation using a limited number of clients may poorly estimate the population covariance $\Sigma_c$. So, we perform shrinkage to better estimate the class covariances from the client means as follows:

$$\hat{\Sigma}_c = \frac{1}{K-1} \sum_{k=1}^{K} n_{k,c} (\hat{\mu}_{k,c} - \hat{\mu}_c)(\hat{\mu}_{k,c} - \hat{\mu}_c)^\top + \gamma I_d, \tag{6}$$

where $\hat{\mu}_{k,c} = \overline{F}_{k,c}$ represents a realization of client means and $\gamma > 0$ is the shrinkage factor.

**Impact of number of clients.** The quality of estimated covariances depends on the number of clients. More clients will give more means and improve the estimate compared to fewer clients. While realistic settings have thousands of clients [18, 20], there can be FL settings with fewer. In such cases, we propose to sample multiple means from each client to increase number of means used for covariance estimation. This can be done by randomly sampling subsets of features in each client and computing a mean from each subset. We validate this approach experimentally (see Figure 5). We discuss more on sampling multiple class means in Appendix E.

## 4.3 Classifier Initialization with Estimated Covariances

Having derived how to compute class covariances from client means, we now describe how FedCOF leverages the estimated class covariances to initialize classifier weights.

**Proposition 2.** *Let $F \in \mathbb{R}^{d \times N}$ be a feature matrix with empirical global mean $\hat{\mu}_g \in \mathbb{R}^d$, and $Y \in \mathbb{R}^{N \times C}$ be a label matrix. The optimal ridge regression solution $W^* = (G + \lambda I_d)^{-1} B$, where $B \in \mathbb{R}^{d \times C}$ and $G \in \mathbb{R}^{d \times d}$ can be written in terms of class means and covariances as follows:*

$$B = [\hat{\mu}_c N_c]_{c=1}^C, \quad G = \sum_{c=1}^C (N_c - 1)\hat{S}_c + \sum_{c=1}^C N_c(\hat{\mu}_c - \hat{\mu}_g)(\hat{\mu}_c - \hat{\mu}_g)^\top + N\hat{\mu}_g\hat{\mu}_g^\top, \quad (7)$$

*where the first two terms $G_{with} = \sum_{c=1}^C (N_c - 1)\hat{S}_c$ and $G_{btw} = \sum_{c=1}^C N_c(\hat{\mu}_c - \hat{\mu}_g)(\hat{\mu}_c - \hat{\mu}_g)^\top$ represents the within-class and between class scatter respectively, while $\hat{\mu}_c$, $\hat{S}_c$ and $N_c$, denote the empirical mean, covariance and sample size for class c, respectively.*

*Proof.* The proof is based on the observation that $G = FF^\top$ from ridge regression is an uncentered and unnormalized empirical global covariance. By using the empirical global covariance definition and decomposing it into within-class and between-class scatter, we obtain the above formulation of $G$. In Appendix D, we provide the detailed proof.

To analyze the impact of the two scatter matrices, we first consider the centralized setting in Table 2 and empirically find that using only within-class scatter matrix performs better than using total scatter matrix in Equation (7). We then analyze their spectral properties via the condition number, defined for a matrix $G$ as $\mathcal{K}(G) = \lambda_{\max}(G)/\lambda_{\min}^+(G)$, where $\lambda_{\max}(G)$ and $\lambda_{\min}^+(G)$ are the largest and smallest non-zero eigenvalue, respectively. For a SqueezeNet backbone in the centralized setting, we observe that $G_{btw}$ is highly ill-conditioned, with condition numbers $\mathcal{K}(G_{btw})$ of $3.0 \times 10^7$, $2.5 \times 10^7$, $2.2 \times 10^7$, $1.3 \times 10^7$ on CUB, Cars, ImageNet-R and CIFAR-100, respectively, while $G_{with}$ is much better conditioned ($\mathcal{K}(G_{with})$: $4.5 \times 10^3$, $2.5 \times 10^4$, $8.2 \times 10^2$, $6.3 \times 10^3$). Including $G_{btw}$ can

Table 2: Analysis showing improved accuracy by removing between-class scatter for classifier initialization in centralized setting using pre-trained SqueezeNet.

| Dataset | $G_{btw}$ | $G_{with}$ | Acc.($\uparrow$) |
|---|---|---|---|
| CIFAR-100 | ✓ | ✓ | 57.1 |
| | ✗ | ✓ | **57.3** |
| ImageNet-R | ✓ | ✓ | 37.6 |
| | ✗ | ✓ | **38.6** |
| CUB200 | ✓ | ✓ | 50.4 |
| | ✗ | ✓ | **53.7** |
| Stanford Cars | ✓ | ✓ | 41.4 |
| | ✗ | ✓ | **44.8** |

cause numerical instability due to its poor conditioning, leading the classifier to overfit directions with small eigenvalues that capture noise or dataset-specific artifacts. This is consistent with the results in Appendix F, where we show that using $G_{btw}$ leads to higher overfitting on the training data.

As a result, we propose to remove the between-class scatter and initialize the linear classifier at the end of the pre-trained network using the within-class covariances $\hat{\Sigma}_c$ which are estimated from client means using Equation (6), as follows:

$$W^* = \hat{G}^{-1}B; \quad \hat{G} = \sum_{c=1}^C (N_c - 1)\hat{\Sigma}_c + N\hat{\mu}_g\hat{\mu}_g^\top. \quad (8)$$

Theoretically, we observe that a similar approach is used in Linear Discriminant Analysis [13], which employs only within-class covariances for finding optimal weights. By removing between-class scatter, we propose a more effective classifier initialization than Fed3R (which uses $G$ from Equation (7) and considers both within- and between-class scatter matrices).

We summarize in Algorithm 1 how we estimate the covariance matrix for each class using only the client means and use the estimated covariances to initialize the classifier as in Equation (8). The normalization of the weights accounts for class imbalance in the entire dataset.

**Impact of the iid assumption**. FedCOF initializes the classifier with the class covariance estimator (Equation 6), unbiased only under the assumption of *iid pre-trained features per class* (Section 4.2). In FL each client has its own data, typically distributed in a statistically heterogeneous or class-imbalanced manner according to a Dirichlet distribution [17]. As a result, each client has data from different set of classes, resulting in non-iid data distributions across clients. However, note that the

**Algorithm 1** FedCOF: Federated Learning with COvariances for Free

| **Client-Side (Client $k$):** | **Server-Side:** |
|---|---|

**Client-Side (Client $k$):**

**Input:** $C$: set of all classes, $f_\theta$: pre-trained model, $X_{k,c}$: samples of class $c$ in client $k$, $n_{k,c}$: number of samples in $X_{k,c}$

**for** $c = 1$ to $C$ **do**
$\quad \hat{\mu}_{k,c} = \frac{1}{n_{k,c}} \sum_{x \in X_{k,c}} f_\theta(x)$
**end for**
**Send** the class means $\hat{\mu}_{k,c}$ and sample counts $n_{k,c}$ to the Server

**Server-Side:**

**Input:** $\hat{\mu}_{k,c}, n_{k,c}$ sent from $K$ clients, $\gamma > 0$
**for** $c = 1 \dots C$ **do**
$\quad \hat{\mu}_c = \frac{1}{N_c} \sum_{k=1}^{K} n_{k,c} \hat{\mu}_{k,c}; N_c = \sum_{k=1}^{K} n_{k,c}$
$\quad \hat{\Sigma}_c = \frac{1}{K-1} \sum_{k=1}^{K} n_{k,c}(\hat{\mu}_{k,c} - \hat{\mu}_c)(\hat{\mu}_{k,c} - \hat{\mu}_c)^\top + \gamma I_d$, Eq.(6)
**end for**
$\hat{\mu}_g = \frac{1}{N} \sum_{c=1}^{C} N_c \hat{\mu}_c, \qquad N = \sum_{c=1}^{C} N_c$
$B = [\hat{\mu}_c N_c]_{c=1}^{C}, \qquad \hat{G} = \sum_{c=1}^{C} (N_c - 1)\hat{\Sigma}_c + N\hat{\mu}_g\hat{\mu}_g^\top$
$W^* = \hat{G}^{-1}B$, Eq. (8)
Normalize $W^*$: $W_c^* \leftarrow W_c^*/\|W_c^*\| \quad c = 1, \dots, C$

---

samples belonging to the same class in different clients are sampled from the same distribution. We exploit this fact in FedCOF. We later show empirically that our method can be successfully applied to non-iid FL scenarios involving thousands of heterogeneous clients on iNaturalist-Users-120K [18]. We hypothesize that this can be attributed to the strong generalization capabilities of pre-trained models. We analyze the bias of the estimator under non-iid assumptions for the same class and also evaluate FedCOF in *feature shift* settings [33] in Appendix G.

**FedCOF in multiple rounds.** While the proposed estimator requires class means from all clients in a single round, this might not be realistic in settings in which clients appear in successive rounds based on availability. For multi-round classifier initialization (see FedCOF in Figure 3 before fine-tuning), the server uses all class means and counts received from all clients seen up to the current round and stores the accumulated means and counts for future rounds. As a result, FedCOF uses statistics from all clients seen up to the current round, similar to Fed3R. This can be easily implemented by ensuring on the client side that each client transfers statistics to the server only once, avoiding repeated transfer of statistics. FedCOF converges when all clients are seen at least once, with total communication cost equal to the single round-case (see Appendix H for details on communication cost).

**Privacy aspects.** FedCOF requires each client to send its class means and frequencies (see Eq. 5), similar to CCVR [35]. While this avoids sharing feature-level data, it does not guarantee strong privacy and may introduce concerns, particularly because it complicates the secure aggregation protocol applicable in Fed3R. Nonetheless, as discussed in Appendix I, FedCOF can be combined with additional privacy-preserving mechanisms, such as adding noise to the shared statistics or adapting the method to support secure aggregation. The latter preserves FedCOF's performance but increases communication overhead to the level of Fed3R.

## 5 Experiments

**Datasets.** We evaluate FedCOF on multiple datasets namely CIFAR-100 [28], ImageNet-R [16] (IN-R), CUB200 [50], Stanford Cars [27] and iNaturalist [48]. We distribute the first 4 datasets to 100 clients using a highly heterogeneous Dirichlet distribution ($\alpha = 0.1$) following standard practice [17, 29]. We also use real-world non-iid FL benchmark of iNaturalist-Users-120K [18] (iNat-120K) having 1203 classes across 9275 clients. We discuss the dataset details in Appendix J.

**Implementation details.** We use three models: namely SqueezeNet [19] following [29] and [38], MobileNetV2 [44] following [11, 18], and ViT-B/16 [9]. All models are pre-trained on ImageNet-1k [7]. We use the FLSim library. We use $\gamma = 1$ for all experiments with SqueezeNet and ViT-B/16, and $\gamma = 0.1$ for all experiments with MobileNetV2 due to very high dimensionality $d$ of the feature space. We compare to *FedCOF Oracle* in which real class covariances are shared from clients and aggregated in server instead of using our estimated covariances (see Appendix J). For all experiments, we set the client participation in each round to 30%, and we show the training-free methods in multiple rounds in Figures 3 and 4. We provide more implementation details in Appendix J. We discuss computation of communication costs for all methods in Appendix H.

**Evaluation for different training-free methods.** We compare the performance of existing training-free methods and the proposed method in Table 3 using pre-trained Squeezenet, Mobilenetv2 and ViT-B/16 models. We observe that Fed3R [11] using second-order statistics outperforms FedNCM [29]

Table 3: Evaluation of different *training-free methods* using 100 clients for first four datasets and 9275 pre-defined clients on iNat-120K across 5 random seeds. We show the total communication cost (in MB) from all clients to server. We also show the FedCOF oracle in which full class covariances are shared from clients to server. Best results from each section in **bold**.

| | Method | SqueezeNet ($d = 512$) Acc (↑) | Comm. (↓) | MobileNetv2 ($d = 1280$) Acc (↑) | Comm. (↓) | ViT-B/16 ($d = 768$) Acc (↑) | Comm. (↓) |
|---|---|---|---|---|---|---|---|
| **CIFAR-100** | FedNCM [29] | 41.5±0.1 | **5.9** | 55.6±0.1 | **14.8** | 55.2±0.1 | **8.9** |
| | Fed3R [11] | **56.9**±0.1 | 110.2 | 62.7±0.1 | 670.1 | **73.9**±0.1 | 244.8 |
| | FedCOF (Ours) | 56.1±0.2 | **5.9** | **63.5**±0.1 | **14.8** | 73.2±0.1 | **8.9** |
| | FedCOF Oracle (Full Covs) | 56.4±0.1 | 3015.3 | 63.9±0.1 | 18823.5 | 73.8±0.1 | 6780.0 |
| **IN-R** | FedNCM [29] | 23.8±0.1 | **7.1** | 37.6±0.2 | **17.8** | 32.3±0.1 | **10.7** |
| | Fed3R [11] | 37.6±0.2 | 111.9 | 46.0±0.3 | 673.1 | **51.9**±0.2 | 246.6 |
| | FedCOF (Ours) | **37.8**±0.4 | **7.1** | **47.4**±0.1 | **17.8** | 51.8±0.3 | **10.7** |
| | FedCOF Oracle (Full Covs) | 38.2±0.1 | 3645.7 | 48.0±0.3 | 22758.8 | 52.7±0.1 | 8197.4 |
| **CUB200** | FedNCM [29] | 37.8±0.3 | **4.8** | 58.3±0.3 | **12.0** | 75.7±0.1 | **7.2** |
| | Fed3R [11] | 50.4±0.3 | 109.6 | 58.6±0.2 | 667.3 | 77.7±0.1 | 243.1 |
| | FedCOF (Ours) | **53.7**±0.3 | **4.8** | **62.5**±0.4 | **12.0** | **79.4**±0.2 | **7.2** |
| | FedCOF Oracle (Full Covs) | 54.4±0.1 | 2472.1 | 63.1±0.5 | 15432.7 | 79.6±0.2 | 5558.6 |
| **Cars** | FedNCM [29] | 19.8±0.2 | **5.4** | 30.0±0.1 | **13.5** | 26.2±0.4 | **8.1** |
| | Fed3R [11] | 39.9±0.2 | 110.2 | 41.6±0.1 | 668.8 | 47.9±0.3 | 244.0 |
| | FedCOF (Ours) | **44.0**±0.3 | **5.4** | **47.3**±0.5 | **13.5** | **52.5**±0.3 | **8.1** |
| | FedCOF Oracle (Full Covs) | 44.6±0.1 | 2767.3 | 47.2±0.3 | 17275.7 | 53.1±0.1 | 6222.5 |
| **iNat-120K** | FedNCM [29] | 21.2±0.1 | **111.8** | 36.0±0.1 | **279.5** | 53.9±0.1 | **167.7** |
| | Fed3R [11] | 32.1±0.1 | 9837.3 | 41.5±0.1 | 61064.1 | 62.5±0.1 | 22050.2 |
| | FedCOF (Ours) | **32.5**±0.1 | **111.8** | **44.1**±0.1 | **279.5** | **63.1**±0.1 | **167.7** |
| | FedCOF Oracle (Full Covs) | 32.4±0.1 | 57k | 43.6±0.1 | 358k | 62.9±0.1 | 128k |

significantly ranging from 0.3% to 21% across all datasets. However, Fed3R requires a higher communication cost compared to FedNCM. In real-world iNat-120K benchmark, Fed3R needs $61k$ MB compared to 280 MB for FedNCM (see Figure 1), which is 218 times higher. FedCOF performs better than Fed3R in most settings despite having the same communication cost as FedNCM. FedCOF achieves similar performance as the oracle setting using aggregated class covariances requiring very high communication, which validates the effectiveness of the proposed covariance estimator.

FedCOF maintains similar accuracy with Fed3R on CIFAR-100 and ImageNet-R, with an improvement of about 1% when using MobileNetv2. FedCOF outperforms Fed3R on CUB200 and Cars. On CUB200, FedCOF outperforms Fed3R by 3.3%, 3.9% and 2.2% using SqueezeNet, MobileNetv2 and ViT-B/16 respectively. FedCOF improves over Fed3R in the range of 4.1% to 5.7% on Cars. On iNat-120K, FedCOF improves over Fed3R by 0.4%, 2.6% and 0.6% using different models. When comparing FedCOF with FedNCM – both with equal communication costs and same strategy in clients – one can observe that the usage of second order statistics derived only from the class means of clients leads to large performance gains, e.g. 24% using SqueezeNet and 26% using ViT-B/16 on Cars, above 8% using all architectures on large-scale iNat-120K. We also adapt CCVR [35] for classifier initialization in Appendix K and show that FedCOF outperforms CCVR.

**Comparison with communication-efficient methods.** We consider the recent Probabilistic Federated Prompt-Tuning (PFPT) approach that aggregates trainable prompt parameters from clients and tunes them in a federated manner while keeping the backbone fixed [54]. We compare PFPT, FedAvg-PT, and FedProx-PT (prompt-tuning variants of FedAvg and FedProx) with the proposed FedCOF. Similar to PFPT [54], we use a pre-trained ViT-B/32, assign class samples to clients using a Dirichlet distribution ($\alpha = 0.1$), and use 3 random seeds. We use the same training hyperparameters as PFPT. We show in Table 4 that FedCOF is much more communication efficient compared to PFPT and achieves higher performance without any training. We also observe that prompt-tuning methods perform poorly on fine-grained datasets. We discuss more on this in Appendix H.

Table 4: Comparison of FedCOF with federated prompt-tuning methods using ViT-B/32.

| Method | CIFAR-100 Acc (↑) | Comm. (↓) | IN-R Acc (↑) | Comm. (↓) | CUB200 Acc (↑) | Comm. (↓) | Cars Acc (↑) | Comm. (↓) |
|---|---|---|---|---|---|---|---|---|
| FedAvg-PT [54] | 74.5±0.5 | 884.7 | 47.6±1.3 | 1622.0 | 37.0±2.0 | 1622.0 | 13.6±1.7 | 1592.5 |
| FedProx-PT [54] | 73.6±0.4 | 884.7 | 47.9±0.5 | 1622.0 | 38.5±0.8 | 1622.0 | 13.7±1.5 | 1592.5 |
| PFPT [54] | 75.1±0.5 | 846.5 | 50.7±0.2 | 1794.4 | 38.6±0.9 | 1765.5 | 12.9±1.1 | 1736.1 |
| FedCOF (Ours) | **75.3**±0.1 | **8.9** | **54.9**±0.2 | **10.7** | **65.0**±0.1 | **7.2** | **50.4**±0.1 | **8.1** |

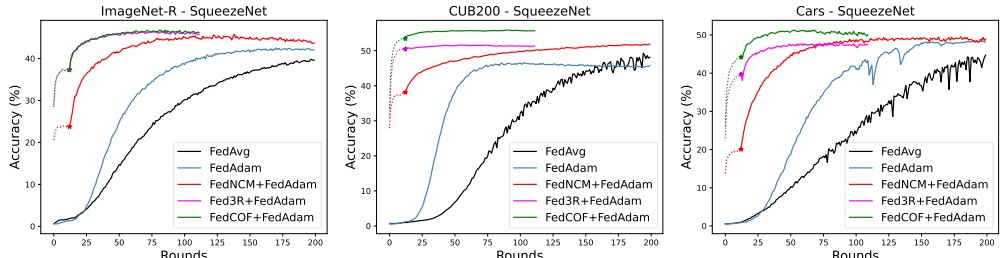

Figure 3: Performance comparison when initializing with different methods and fine-tuning with FedAdam [42] and FedAvg [37]. We also compare with FedAdam and FedAvg using a pre-trained backbone and random classifier initialization. The training-free initialization stages are shown as dotted lines and stars represents the start of fine-tuning stages. Accuracies are averaged over 3 seeds.

**Comparison with full fine-tuning methods.** We compare training-free methods with FL baselines like FedAvg and FedAdam with randomly initialized classifier and pre-trained backbone in Table 5. We use adaptive optimizer, FedAdam [42] since it performs better than most other optimizers as shown in [38]. Without any training, FedCOF outperforms FedAvg in all settings and FedAdam by 7.3% on CUB200 and 2.2% on Cars, and achieves competitive performance in ImageNet-R. We show in Figure 3 how FedCOF starts from a very high accuracy compared to FedAdam and further improves on fine-tuning. We provide more experiments with pre-trained ResNet-18 in Appendix K.

**Analysis of fine-tuning and linear probing.** While FedCOF achieves high accuracy without training, we show in Figure 3 that further fine-tuning the model with FL achieves better and faster convergence compared to federated optimization from scratch. We show the performance of fine-tuning after training-free classifier initialization in Figure 3. These training-free methods end after all clients appear at least once to share their local statistics. We fine-tune the models after FedCOF and Fed3R for 100 rounds since they achieve fast convergence, while we train for 200 rounds for

Table 5: Comparison with training-based FL baselines (FedAvg and FedAdam) using pre-trained SqueezeNet. For training-based methods, we consider 100 rounds of training for fair comparison and report accuracy of 3 random seeds.

| Method | Training | ImageNet-R | CUB200 | Cars |
|---|---|---|---|---|
| FedAvg | ✓ | 30.0±0.6 | 30.3±6.7 | 24.9±1.6 |
| FedAdam | ✓ | 38.8±0.6 | 46.4±0.8 | 41.8±0.6 |
| FedNCM | ✗ | 23.8±0.1 | 37.8±0.3 | 19.8±0.2 |
| Fed3R | ✗ | 37.6±0.2 | 50.4±0.3 | 39.9±0.2 |
| FedCOF (Ours) | ✗ | 37.8±0.4 | 53.7±0.3 | 44.0±0.3 |
| FedNCM+FedAdam | ✓ | 44.7±0.1 | 50.2±0.2 | 48.7±0.2 |
| Fed3R+FedAdam | ✓ | 45.9±0.3 | 51.2±0.3 | 47.4±0.4 |
| FedCOF+FedAdam | ✓ | **46.0**±0.4 | **55.7**±0.4 | **49.6**±0.6 |

FedAdam, FedAvg, and fine-tuning after FedNCM which takes longer to converge. Fine-tuning after FedCOF starts at a higher accuracy and converges faster compared to FedNCM. Although FedCOF and Fed3R converge similarly on ImageNet-R, FedCOF+FedAdam achieves better accuracy than Fed3R+FedAdam on CUB200 and Cars. We see in Table 5 that all training-free approaches followed by fine-tuning outperform FedAdam and FedAvg with random classifier initialization.

Following [29] and [38], we perform federated linear probing (LP) of the models using FedAvg after classifier initialization with training-free methods. In FedAvg-LP, we perform FedAvg and learn only the classifier weights of all client models. Linear probing requires much less computation compared to fine-tuning the entire model and were found to be effective with pre-trained models. We observe in Figure 4 that linear probing after FedCOF improves significantly compared to FedNCM and Fed3R using ViT-B/16 on Cars and SqueezeNet on iNat-120K. On the real-world dataset iNat-120K, FedAvg-LP with random classifier initialization achieves 27.3% after 5000 rounds while FedCOF+FedAvg-Lp achieves 34% in less than 1000 rounds. We plot accuracy versus communication in Figure 4 (center) to demonstrate the advantage of FedCOF over other methods. We discuss more in Appendix K.

**Impact of number of clients and data heterogeneity.** We analyze in Figure 5 (left), the performance of FedCOF with varying number of clients and data heterogeneity. We observe that the performance of FedCOF improves with increasing number of clients and decreasing heterogeneity. This is due to the fact that more clients provides more class means and more uniform data distribution gives better representative local means. While more clients are favourable for FedCOF, it still performs well and outperforms FedNCM significantly in the setting with 10 clients and high data heterogeneity.

**Multiple class means per client.** We analyze FL settings with fewer clients ranging from 10 to 50 in Figure 5 (center) and show that sharing multiple class means from each client improves the accuracy. Using only 10 clients, sharing 2 class means per client improves the accuracy by 2.6%.

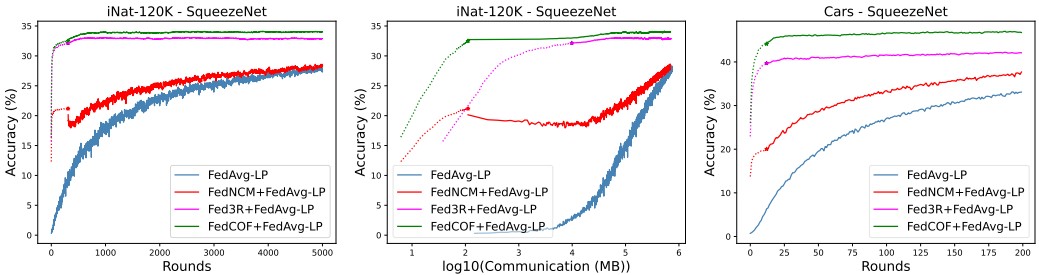

Figure 4: Performance of different classifier initialization methods when linear-probing with Fe-dAvg [37]. FedAvg-LP (in blue) uses random classifier initialization and a pre-trained backbone. The training-free initialization stage is shown in dotted lines, stars represents the start of linear probing.

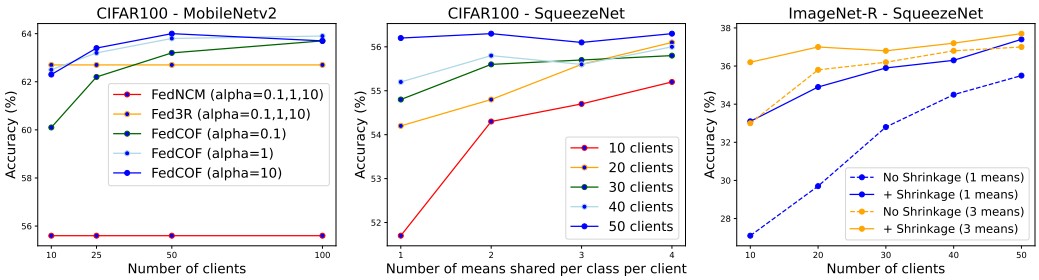

Figure 5: Ablation experiments: (left) Change in performance with varying number of clients and data heterogeneity. (center) Sharing multiple class means per client improves FedCOF performance. (right) Impact of shrinkage with varying number of clients and sampled means per client.

**Impact of shrinkage.** We show in Figure 5 (right) that using shrinkage improves the covariance estimates thereby improving the accuracy. We observe that shrinkage consistently improves performance, especially when the number of sampled means is small. When the number of means is low relative to the feature dimension ($d = 512$ in SqueezeNet), the covariance estimate becomes ill-conditioned, and shrinkage stabilizes it. However, as more class means are sampled per client or the number of clients increases, the benefit of shrinkage diminishes, since the total number of means approaches the feature dimension, making the estimate more stable. We present more ablation studies in Appendix L.

## 6 Conclusion

In this work we proposed FedCOF, a novel training-free approach for federated learning with pre-trained models. By leveraging the statistical properties of client class sample means, we showed that second-order statistics can be estimated using only class means from clients, thus reducing communication costs. We derived a provably unbiased estimator of population class covariances, enabling accurate estimation of a global covariance matrix. By applying shrinkage to the estimated class covariances and removing between-class scatter matrices, the server can effectively use this global covariance to initialize a global classifier. Our experiments demonstrated that FedCOF outperforms FedNCM [29] by significant margins while maintaining the same communication cost. Additionally, FedCOF delivers competitive or even superior results to Fed3R [11] across model architectures and benchmarks while substantially reducing communication costs. Finally, we showed that FedCOF outperforms federated prompt-tuning methods and serves as a more effective starting point for improving the convergence of federated fine-tuning and linear probing methods.

**Limitations.** The quality of our estimator depends on the number of clients, as shown in Figure 5 where using multiple class means per client helps with fewer clients. Another limitation is the assumption that samples of the same class are iid across clients, which is, however, an assumption underlying most of federated learning. We discuss the bias in our estimator in non-iid settings in Appendix G.

**Acknowledgements.** We acknowledge project PID2022-143257NB-I00, financed by MCIN/AEI/10.13039/501100011033 and ERDF/EU and FSE+, funding by the European Union ELLIOT project, and the Generalitat de Catalunya CERCA Program. Bartłomiej Twardowski acknowledges the grant RYC2021-032765-I and National Centre of Science (NCN, Poland) Grant No. 2023/51/D/ST6/02846. Kai Wang acknowledges the funding from Guangdong and Hong Kong Universities 1+1+1 Joint Research Collaboration Scheme and the start-up grant B01040000108 from CityU-DG. Andrew Bagdanov acknowledges financial support from the AVIOS project, financed by the Italian Ministry of Enterprises and Made in Italy (MiSE/MIMIT). We acknowledge all reviewers, especially anonymous reviewer (Md8N) whose constructive feedback restored our belief in the review system, which had been severely shaken in earlier review rounds.

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

# Appendix

## A  Scope and Summary of Notation

These appendices provide additional information, proofs, experimental results, and analyses that complement the main paper. For clarity and convenience, here we first summarize the key notations used throughout the paper:

- $N$: total number of samples.
- $K$: number of clients.
- $C$: number of classes.
- $d$: dimensionality of the feature space.
- $n_{k,c}$: number of samples from class $c$ assigned to client $k$.
- $N_c = \sum_{k=1}^{K} n_{k,c}$: total number of samples in class $c$.
- $\hat{\mu}_g, \hat{\mu}_c \in \mathbb{R}^d$: *empirical* global mean and class mean for class $c$, respectively.
- $\mu_c \in \mathbb{R}^d$: *population* mean of class $c$.
- $\hat{S}_c \in \mathbb{R}^{d \times d}$: *empirical* sample covariance for class $c$.
- $\Sigma_c \in \mathbb{R}^{d \times d}$: *population* covariance for class $c$.
- $\hat{\Sigma}_c \in \mathbb{R}^{d \times d}$: our unbiased estimator of the population covariance $\Sigma_c$ employing only client means.
- $F \in \mathbb{R}^{d \times N}$: feature matrix, where each column $F^j \in \mathbb{R}^d$ is a feature vector, for $j = 1, \ldots, N$.
- $F_{k,c}^j \in \mathbb{R}^d$: $j$-th feature vector from class $c$ assigned to client $k$.
- $\overline{F}_{k,c} \in \mathbb{R}^d$: sample mean of the feature vectors for class $c$ on client $k$, treated as a random vector. A specific realization of this random vector is denoted by $\hat{\mu}_{k,c}$.
- $\mathrm{Var}[\overline{F}_{k,c}] = \mathrm{Cov}[\overline{F}_{k,c}, \overline{F}_{k,c}]$ represents the covariance matrix of the random vector $\overline{F}_{k,c}$.

## B  Expectation and Variance of the Sample Mean

Let $\{F_{k,c}^j\}_{j=1}^{n_{k,c}}$ be a random sample from a multivariate population with mean $\mu_c$ and covariance $\Sigma_c$, where $F_{k,c}^j$ is the $j$-$th$ feature vector of class $c$ assigned to the client $k$ and $n_{k,c}$ is the number of elements of class $c$ in the client $k$. Assuming that the per-class features $F_{k,c}^j$ in each client are iid in the initialization, then the sample mean of the features for class $c$

$$\overline{F}_{k,c} = \frac{1}{n_{k,c}} \sum_{j=1}^{n_{k,c}} F_{k,c}^j, \tag{9}$$

is distributed with mean and covariance given by:

$$\mathbb{E}[\overline{F}_{k,c}] = \mu_c \qquad \mathrm{Var}[\overline{F}_{k,c}] = \frac{\Sigma_c}{n_{k,c}} \tag{10}$$

*Proof.* To prove this, we fix the class $c$ and omit the dependencies on $c$ for simplicity. Thus, we write $n_{k,c} = n_k$, $F_{k,c}^j = F_k^j$, $\overline{F}_{k,c} = \overline{F}_k$, and $\mu_c = \mu$, $\Sigma_c = \Sigma$.

Since $\{F_k^j\}_{j=1}^{n_k}$ is a random sample from a multivariate distribution with mean $\mu$ and covariance $\Sigma$, and the per-class features $F_k^j$ in each client are i.i.d at initialization, it follows that:

$$\mathbb{E}[F_k^j] = \mu \qquad \mathrm{Var}[F_k^j] = \Sigma, \quad \forall j \tag{11}$$

By computing the expectation of $\overline{F}_k$ and using the linearity of expectation, we obtain:

$$\mathbb{E}[\overline{F}_k] = \mathbb{E}[\frac{1}{n_k} \sum_{j=1}^{n_k} F_k^j] = \frac{1}{n_k}\mathbb{E}[F_k^1] + \ldots + \frac{1}{n_k}\mathbb{E}[F_k^{n_k}] = \frac{1}{n_k}(n_k\mu) = \mu,$$

where in the last equality we used Equation (11). Thus the expectation of the sample mean is $\mu$, which completes the first part of the proof.

Next, we show that the variance of the sample mean is $\frac{\Sigma}{n_k}$. By computing the variance of $\overline{F}_k$ and using the fact that the variance scales by the square of the constant, we obtain:

$$\mathrm{Var}[\overline{F}_k] = \mathrm{Var}[\frac{1}{n_k}\sum_{j=1}^{n_k} F_k^j] = \frac{1}{n_k^2}\left(\mathrm{Var}[F_k^1] + \ldots + \mathrm{Var}[F_k^{n_k}]\right) + \frac{1}{n_k^2}\sum_{i=1}^{n_k}\sum_{\substack{j=1\\j\neq i}}^{n_k}\mathrm{Cov}[F_k^i, F_k^j].$$

By the independence assumption of $\{F_k^j\}_{j=1}^{n_k}$, the cross terms $\mathrm{Cov}[F_k^i, F_k^j] = 0$ for $i \neq j$. Applying Equation (11), we have:

$$\mathrm{Var}[\overline{F}_k] = \frac{1}{n_k^2}\left(\mathrm{Var}[F_k^1] + \ldots + \mathrm{Var}[F_k^{n_k}]\right) = \frac{1}{n_k^2}(n_k\Sigma) = \frac{\Sigma}{n_k}$$

$\square$

## C   Proof of Proposition 1

**Proposition 1.** *Let $K$ be the number of clients, each with $n_{k,c}$ features, and let $C$ be the total number of classes. Let $\hat{\mu}_c = \frac{1}{N_c}\sum_{j=1}^{N_c} F^j$ be the unbiased estimator of the population mean $\mu_c$ and $N_c = \sum_{k=1}^{K} n_{k,c}$ be the total number of features for a single class. Assuming the features for class $c$ are iid across clients at initialization, the estimator*

$$\hat{\Sigma}_c = \frac{1}{K-1}\sum_{k=1}^{K} n_{k,c}(\overline{F}_{k,c} - \hat{\mu}_c)(\overline{F}_{k,c} - \hat{\mu}_c)^\top \tag{12}$$

*is an unbiased estimator of the population covariance $\Sigma_c$, for all $c \in 1, \ldots, C$.*

*Proof.* To prove this, we fix the class $c$ and omit the dependencies on $c$ for clarity. So we write $n_{k,c} = n_k$, $\overline{F}_{k,c} = \overline{F}_k$, $N_c = N$, $\hat{\mu}_c = \hat{\mu}$, $\hat{\Sigma}_c = \hat{\Sigma}$, $\mu_c = \mu$, and $\Sigma_c = \Sigma$. By the definition of an unbiased estimator, we need to show that:

$$\mathbb{E}[\hat{\Sigma}] = \mathbb{E}\left[\frac{1}{K-1}\sum_{k=1}^{K} n_k(\overline{F}_k - \hat{\mu})(\overline{F}_k - \hat{\mu})^\top\right] = \Sigma.$$

By the linearity of the expectation, the definition of sample mean $\overline{F}_k = \frac{1}{n_k}\sum_{j=1}^{n_k} F_k^j$, and the definition of global class mean $\hat{\mu} = \frac{1}{N}\sum_{k=1}^{K}\sum_{j=1}^{n_k} F_k^j$, we have:

$$\mathbb{E}[\hat{\Sigma}] = \frac{1}{K-1}\left(\sum_{k=1}^{K} n_k\mathbb{E}[\overline{F}_k\overline{F}_k^\top] - \sum_{k=1}^{K} n_k\mathbb{E}[\overline{F}_k\hat{\mu}^\top] - \sum_{k=1}^{K} n_k\mathbb{E}[\hat{\mu}\overline{F}_k^\top] + \sum_{k=1}^{K} n_k\mathbb{E}[\hat{\mu}\hat{\mu}^\top]\right)$$

$$= \frac{1}{K-1}\left(\sum_{k=1}^{K} n_k\mathbb{E}[\overline{F}_k\overline{F}_k^\top] - 2\mathbb{E}[(\sum_{k=1}^{K}\sum_{j=1}^{n_k} F_k^j)\hat{\mu}^\top] + \sum_{k=1}^{K} n_k\mathbb{E}[\hat{\mu}\hat{\mu}^\top]\right)$$

$$= \frac{1}{K-1}\left(\sum_{k=1}^{K} n_k\mathbb{E}[\overline{F}_k\overline{F}_k^\top] - 2N\mathbb{E}[\hat{\mu}\hat{\mu}^\top] + \sum_{k=1}^{K} n_k\mathbb{E}[\hat{\mu}\hat{\mu}^\top]\right). \tag{13}$$

By applying the variance definition, along with the expectation and variance of the sample mean (see Equation (10)), we obtain:

$$\mathbb{E}[\overline{F}_k\overline{F}_k^\top] = \mathrm{Var}[\overline{F}_k] + \mathbb{E}[\overline{F}_k]\mathbb{E}[\overline{F}_k]^\top = \frac{\Sigma}{n_k} + \mu\mu^\top. \tag{14}$$

Now, by considering the right term in Equation (13), since $\hat{\mu}$ is an unbiased estimator of the population mean, then $\mathbb{E}[\hat{\mu}] = \mu$. Moreover, since we assume that the features for a single class across clients are i.i.d at initialization, we can use re-use the result in Equation (10) by considering the all class features as a random sample of size $N$ from a population with mean $\mu$ and variance $\Sigma$. Consequently, the global sample mean $\hat{\mu}$ is has variance $\mathrm{Var}[\hat{\mu}] = \frac{\Sigma}{N}$. Then

$$\mathbb{E}[\hat{\mu}\hat{\mu}^\top] = \mathrm{Var}[\hat{\mu}] + \mathbb{E}[\hat{\mu}]\mathbb{E}[\hat{\mu}]^\top = \frac{\Sigma}{N} + \mu\mu^\top. \tag{15}$$

By using Equation (14) and Equation (15) in Equation (13), and recalling that $N = \sum_{k=1}^K n_k$, we obtain:

$$\mathbb{E}[\hat{\Sigma}] = \frac{1}{K-1}\left(\sum_{k=1}^K n_k(\frac{\Sigma}{n_k} + \mu\mu^\top) - 2N(\frac{\Sigma}{N} + \mu\mu^\top) + \sum_{k=1}^K n_k(\frac{\Sigma}{N} + \mu\mu^\top)\right)$$

$$= \frac{1}{K-1}(K\Sigma + \mu\mu^\top N - 2\Sigma - 2N\mu\mu^\top + (\frac{\Sigma}{N} + \mu\mu^\top)N) = \frac{1}{K-1}(K-1)\Sigma = \Sigma.$$

$\square$

## D  Proof of Proposition 2

**Proposition 2.** *Let $F \in \mathbb{R}^{d \times N}$ be a feature matrix with empirical global mean $\hat{\mu}_g \in \mathbb{R}^d$, and $Y \in \mathbb{R}^{N \times C}$ be a label matrix. The optimal ridge regression solution $W^* = (G + \lambda I_d)^{-1}B$, where $B \in \mathbb{R}^{d \times C}$ and $G \in \mathbb{R}^{d \times d}$ can be written in terms of class means and covariances as follows:*

$$B = [\hat{\mu}_c N_c]_{c=1}^C, \tag{16}$$

$$G = \sum_{c=1}^C (N_c - 1)\hat{S}_c + \sum_{c=1}^C N_c(\hat{\mu}_c - \hat{\mu}_g)(\hat{\mu}_c - \hat{\mu}_g)^\top + N\hat{\mu}_g\hat{\mu}_g^\top \tag{17}$$

*where the first two terms $\sum_{c=1}^C (N_c - 1)\hat{S}_c$ and $\sum_{c=1}^C N_c(\hat{\mu}_c - \hat{\mu}_g)(\hat{\mu}_c - \hat{\mu}_g)^\top$ represents the within-class and between class scatter respectively, while $\hat{\mu}_c$, $\hat{S}_c$ and $N_c$, denote the empirical mean, covariance and sample size for class c, respectively.*

*Proof.* The first part, regarding Equation (16), follows directly. From the ridge regression solution, $B = FY$, which is obtained by summing the features for each class and arranging them into the columns of a matrix. This results in the product of class means and samples per class.

Now, for computing the matrix $G$, we proceed with the definition of the global sample covariance:

$$\hat{S} = \frac{1}{N-1}(F - \overline{F})(F - \overline{F})^\top = \frac{1}{N-1}\left(FF^\top - F\overline{F}^\top - \overline{F}F^\top + \overline{F}\,\overline{F}^\top\right),$$

where $\overline{F} = \left(\frac{1}{N}\sum_{j=1}^N F^j\right)\mathbf{1}^\top = \hat{\mu}_g\mathbf{1}^\top \in \mathbb{R}^{d \times N}$ is the matrix obtained by replicating the global mean $N$ times in each column and $\mathbf{1} \in \mathbb{R}^{N \times 1}$ is a column vector of ones. Recalling that $G = FF^\top$, we have:

$$\hat{S} = \frac{1}{N-1}(G - F\mathbf{1}\hat{\mu}_g^\top - \hat{\mu}_g\mathbf{1}^\top F^\top + \hat{\mu}_g\mathbf{1}^\top\mathbf{1}\hat{\mu}_g^\top) = \frac{1}{N-1}(G - 2F\mathbf{1}\hat{\mu}_g^\top + N\hat{\mu}_g\hat{\mu}_g^\top)$$

since $F\mathbf{1}\hat{\mu}_g^\top = \hat{\mu}_g\mathbf{1}^T F^\top$ and $\mathbf{1}^T\mathbf{1} = N$.

Now, since $F\mathbf{1} = \sum_{j=1}^N F^j$, we can obtain the matrix $G$ as:

$$G = (N-1)\hat{S} + 2\left(\sum_{j=1}^N F^j\right)\hat{\mu}_g^\top - N\hat{\mu}_g\hat{\mu}_g^\top = (N-1)\hat{S} + 2N\hat{\mu}_g\hat{\mu}_g^\top - N\hat{\mu}_g\hat{\mu}_g^\top = (N-1)\hat{S} + N\hat{\mu}_g\hat{\mu}_g^\top \tag{18}$$

It is a well known result that the global covariance can be expressed as:

$$\hat{S} = \frac{1}{N-1}\left(\sum_{c=1}^C (N_c - 1)\hat{\Sigma}_c + \sum_{c=1}^C N_c(\hat{\mu}_c - \hat{\mu}_g)(\hat{\mu}_c - \hat{\mu}_g)^T\right),$$

Replacing the global covariance $\hat{S}$ in Equation (18), we obtain the final expression for $G$ as:

$$G = \sum_{c=1}^{C}(N_c - 1)\hat{S}_c + \sum_{c=1}^{C} N_c(\hat{\mu}_c - \hat{\mu}_g)(\hat{\mu}_c - \hat{\mu}_g)^\top + N\hat{\mu}_g\hat{\mu}_g^\top$$

$\square$

## E   Sampling Multiple Class Means per Client

From a theoretical perspective, although the estimator $\hat{\Sigma}_c$ is unbiased in Equation (5), this only ensures that its expected value equals the true covariance $\Sigma_c$ – not that any individual estimate is accurate. Its variance still depends on the number of independent means $K$ sampled – that is the number of clients in the federation $K$ which contains class $c$.

Therefore, increasing $K$ leads to a tighter concentration of $\hat{\Sigma}_c$ around its expectation, reducing the overall mean square error (MSE) – the sum of variance and squared bias (which remains zero since the estimator is unbiased) – between the estimator and the true covariance matrix. Moreover, where the number of clients is small relative to the feature dimension $d$, the estimate may be ill-conditioned. The shrinkage term $\gamma I_d$ in Equation (6) improves numerical stability in these settings, trading a small amount of bias for a significant reduction in variance – even when only a few clients are available.

**Sampling strategy.** We propose using a simple multiple mean sampling strategy following sampling without replacement. We consider the number of means $\mathcal{M}$ to sample as a fixed number which is a hyperparameter. For each class, we take disjoint random sets from $n_{k,c}$ samples in a client and compute the mean for these subsets. We take disjoint sets to avoid computing similar sample means. The only condition we enforce is that clients use atleast 2 samples to compute a mean. If a client does not have atleast $2\mathcal{M}$ samples, we send less than $\mathcal{M}$ sample means. For instance, if a client has 3 samples, we compute and share a single mean.

We observe that the proposed sampling approach improves performance. However, more sophisticated sampling approaches could be employed if the user is interested in improving performance in FL settings with very few clients. One approach could be sampling means based on the number of samples $n_{k,c}$ instead of using a fixed number of means to sample from every client. If a client has more samples, it could share more sample means and this would thus share more means overall and improve the covariance estimates. Future work could optimize this multiple mean sampling approach to better suit fewer client FL settings.

## F   On Excluding Between-class Scatter

Intuitively, to represent the feature distribution of each class we do not really need to consider the relationships between different classes which represent the distribution of the overall dataset since our goal is to estimate the class-specific classifier weights using these covariances. Based on this intuition, we propose to remove the between-class scatter from Equation (7) and initialize the classifier weights using only the respective within-class scatter matrices.

We also analyze in Table 6 using the centralized setting how the different scatter matrices affect overfitting of the model. We observe that, while both methods achieve similarly high training accuracies, Ridge Regression consistently underperforms on the test set. This suggests that incorporating $G_{\text{btw}}$ introduces an overfitting effect, as the classifier learns directions that do not transfer well to unseen samples.

Finally, we also empirically analyze in Table 7 the impact of removing the between-class covariances in a Federated scenario using SqueezeNet. Here in the FL setup we again clearly see the negative effect of incorporating between-class scatter statistics.

## G   Bias of the Estimator with non-iid Client Features

In Appendix C we showed that, under the assumption that the per-class features are iid across clients, the proposed estimator is an *unbiased estimator*. In this section, we theoretically quantify the bias when the i.i.d assumption is violated.

Table 6: Comparison of classifiers across datasets with and without $G_{\text{btw}}$ in the centralized setting

| Classifier | $G_{\text{with}}$ | $G_{\text{btw}}$ | Dataset | Train Acc | Test Acc |
|---|---|---|---|---|---|
| Ridge Regression | ✓ | ✓ | CIFAR-100 | 60.0 | 57.1 |
| Ours | ✓ | ✗ | CIFAR-100 | 60.4 | **57.3** |
| Ridge Regression | ✓ | ✓ | Imagenet-R | 52.8 | 37.6 |
| Ours | ✓ | ✗ | Imagenet-R | 53.4 | **38.6** |
| Ridge Regression | ✓ | ✓ | CUB200 | 92.0 | 50.4 |
| Ours | ✓ | ✗ | CUB200 | 91.3 | **53.7** |
| Ridge Regression | ✓ | ✓ | Cars | 85.9 | 41.4 |
| Ours | ✓ | ✗ | Cars | 86.2 | **44.8** |

Table 7: Performance comparison of different FL setups using $G_{\text{btw}}$ and $G_{\text{with}}$ across datasets.

| FL Setup | CIFAR-100 | ImageNet-R | CUB200 | CARS |
|---|---|---|---|---|
| Using $G_{\text{btw}}$ | 52.8 | 34.8 | 49.7 | 33.7 |
| Using $G_{\text{btw}} + G_{\text{with}}$ | 56.3 | 36.8 | 51.6 | 42.4 |
| Using $G_{\text{with}}$ (Ours) | 56.3 | 37.2 | 53.5 | 44.6 |

Under the i.i.d. assumption, the single class features assigned to clients can be treated as random samples from the *same* population distribution with mean $\mu_c$ and covariance $\Sigma_c$. For simplicity, focusing on a single class and dropping the class subscript $c$, the population distribution has mean $\mu$ and covariance $\Sigma$. As a result, recalling Equation (14), we can write:

$$\mathbb{E}[\overline{F}_k \overline{F}_k^\top] = \mathrm{Var}[\overline{F}_k] + \mathbb{E}[\overline{F}_k]\mathbb{E}[\overline{F}_k]^\top = \frac{\Sigma}{n_k} + \mu\mu^\top,$$

where $n_k$ is the number of samples assigned to client $k$, and $\overline{F}_k$ is the sample mean for client $k$

Now, if the *i.i.d assumption is violated* the local features assigned to each client can be viewed as random samples drawn from different client population distributions, each characterized by a mean $\mu_k$ and covariance $\Sigma_k$, with $\mu_i \neq \mu_j$ and $\Sigma_i \neq \Sigma_j$ for $i \neq j$, and $i, j = 1, \dots, K$. In this case:

$$\mathbb{E}[\overline{F}_k \overline{F}_k^\top] = \mathrm{Var}[\overline{F}_k] + \mathbb{E}[\overline{F}_k]\mathbb{E}[\overline{F}_k]^\top = \frac{\Sigma_k}{n_k} + \mu_k\mu_k^\top. \tag{19}$$

To compute the expectation of the estimator $\mathbb{E}[\hat{\Sigma}]$, we follow the same procedure used to prove proposition in Appendix C up to Equation (13):

$$\mathbb{E}[\hat{\Sigma}] = \frac{1}{K-1}\left(\sum_{k=1}^{K} n_k \mathbb{E}[\overline{F}_k \overline{F}_k^\top] - 2N\mathbb{E}[\hat{\mu}\hat{\mu}^\top] + \sum_{k=1}^{K} n_k \mathbb{E}[\hat{\mu}\hat{\mu}^\top]\right). \tag{20}$$

Assuming the global feature dataset, regardless of client assignment, is a random sample from the population with mean $\mu$ and covariance $\Sigma$, we can write:

$$\mathbb{E}[\hat{\mu}\hat{\mu}^\top] = \mathrm{Var}[\hat{\mu}] + \mathbb{E}[\hat{\mu}]\mathbb{E}[\hat{\mu}]^\top = \frac{\Sigma}{N} + \mu\mu^\top. \tag{21}$$

Substituting Equation (21) and Equation (19) into Equation (20), and recalling that $N = \sum_{k=1}^{K} n_k$, we obtain:

$$\mathbb{E}[\hat{\Sigma}] = \frac{1}{K-1}\left(\sum_{k=1}^{K} n_k(\frac{\Sigma_k}{n_k} + \mu_k\mu_k^\top) - 2N(\frac{\Sigma}{N} + \mu\mu^\top) + \sum_{k=1}^{K} n_k(\frac{\Sigma}{N} + \mu\mu^\top)\right)$$

$$= \frac{1}{K-1}\left(\sum_{k=1}^{K} n_k(\frac{\Sigma_k}{n_k} + \mu_k\mu_k^\top) - \Sigma - N\mu\mu^\top\right)$$

$$= \frac{1}{K-1}\sum_{k=1}^{K}(\Sigma_k - \frac{\Sigma}{K}) + \frac{1}{K-1}\left(\sum_{k=1}^{K} n_k\mu_k\mu_k^\top - \sum_{k=1}^{K} n_k\mu\mu^\top\right)$$

$$= \frac{1}{K-1}\sum_{k=1}^{K}(\Sigma_k - \frac{\Sigma}{K}) + \frac{1}{K-1}\sum_{k=1}^{K} n_k(\mu_k\mu_k^\top - \mu\mu^\top)$$

$$= \frac{1}{K-1}\sum_{k=1}^{K}(\Sigma_k - \frac{\Sigma}{K}) + \frac{1}{K-1}\sum_{k=1}^{K} n_k(\mu_k - \mu)(\mu_k - \mu)^\top,$$

where in the last step we used that $\sum_{k=1}^{K} n_k\mu_k = N\mu$.

The bias of the estimator is thus given by:

$$\text{Bias}(\hat{\Sigma}) = \mathbb{E}[\hat{\Sigma}] - \Sigma = \frac{1}{K-1}\sum_{k=1}^{K}(\Sigma_k - \Sigma) + \frac{1}{K-1}\left(\sum_{k=1}^{K} n_k(\mu_k - \mu)(\mu_k - \mu)^\top\right). \quad (22)$$

Note that if each client population covariance $\Sigma_k$ is equal to the global population covariance $\Sigma$, and the mean of each client $\mu_k$ is equal to the population mean, then the bias is zero (i.e., the estimator is unbiased). However, the bias formula reveals that when the distribution of a class within a client differs from the global distribution of the same class, our estimator introduces a systematic bias. This situation can arise in the *feature-shift* setting, in which each client is characterized by a different domain. We next evaluate FedCOF under the feature-shift setting to quantify how this bias affects performance in this specific scenario.

### G.1  Experiments on Feature Shift Settings.

Following [33], we perform experiments with MobileNetv2 in a non-iid feature shift setting on the DomainNet [40] dataset. DomainNet contains data from six different domains: Clipart, Infograph, Painting, Quickdraw, Real, and Sketch. We use the top 10 most common classes of DomainNet for our experiments following the setting proposed by [33]. We consider six clients where each client has i.i.d. data from one of the six domains. As a result, different clients have data from different feature distributions.

Table 8: Comparison of different training-free methods using MobileNetV2 on the feature shift setting on DomainNet. We show the total communication cost (in MB) from all clients to server.

| Method | Acc (↑) | Comm. (↓) |
|---|---|---|
| FedNCM | 65.8 | 0.3 |
| Fed3R | 81.9 | 39.6 |
| FedCOF | 74.1 | 0.3 |
| FedCOF (2 class means per client) | 76.5 | 0.6 |
| FedCOF (10 class means per client) | 78.8 | 3.1 |

We show in Table 8 how training-free methods perform in feature shift settings and the accuracy to communication trade-offs.

Fed3R achieves better overall performance then FedCOF, likely due to its use of exact class covariance, avoiding the bias that FedCOF introduces. However, FedCOF achieves comparable results while significantly reducing communication costs. FedNCM perform worse than FedCOF at the same communication budget. When we increase the number of means sampled from each client, the performance of our approach improves. This is due to the fact that our method suffers with low number of clients (only 6 in this experiments) and sampling multiple means helps.

While non-iid feature shift settings have been studied in some papers not using pre-trained models, using this setting with pre-trained models works a bit differently. When using a pre-trained model, the generalization capabilities of the pre-trained model can help in moving the distribution of class features across clients towards an iid feature distribution even if the class distribution across clients

is non-iid at the image level. We believe that more comprehensive analysis of feature-shift settings when using pre-trained models requires more extensive benchmarks with higher number of clients and could be an interesting direction to explore in future works.

# H  Communication Costs

When computing communication costs we consider that the pre-trained models are on the clients similar to [11] and do not need to be communicated. All parameters are considered to be 32-bit floating point numbers (i.e. 4 bytes) in all our analyses and experiments.

For *training-free* methods, each client sends its local statistics to the server only once, as in [29, 11]. In the multi-round federated learning setting, this can be efficiently implemented by ensuring on the client-side that each client only shares its local statistics during its first participation round. This avoids repeated communication of statistics from clients for all training-free methods. Following [11], the communication from server to client is not considered since the clients does not need to receive any updates from the server as the client models are not updated in the training-free initialization of the global classifier. Thus, the communication cost is the same for a single round setting with full client participation and the setting with multiple rounds having partial client participation in each round.

Let $C_k$ denote the number of classes present in client $k$. Due to the non-iid Dirichlet data distribution, $C_k < C$, where $C$ is the total number of classes in the dataset. As a result, the communication cost varies across clients. Defining $M = \sum_{k=1}^{K} C_k$ as the total number of class means shared from the $K$ clients and assuming that each class mean is a $d$-dimensional feature vector, the communication cost of each evaluated approach is given by:

- FedNCM shares a total of $M$ means from $K$ clients. Cost = $Md \cdot 32$.
- Fed3R shares a total of $M$ sum of class features and a total of $K$ matrices of $d^2$ dimensionality from $K$ clients. Cost = $(Md + Kd^2) \cdot 32$.
- FedCOF shares a total of $M$ means from $K$ clients. Cost = $Md \cdot 32$.
- CCVR and FedCOF-Oracle shares a total of $M$ means and $M$ covariance matrices of $d^2$ dimensionality from $K$ clients. Cost = $M \cdot (d + d^2) \cdot 32$.

For the non-iid sampling in experiments reported in Table 3, the values of $M$ for CIFAR-100, ImageNet-R, CUB200, Cars and iNat-120K are 2870, 3470, 2353, 2634 and 54590, respectively.

Instead, for *federated finetuning*, the communication cost is calculated as follows. Let $E$, $C$ and $H = d \cdot C$ denote the sizes of the feature extractor, the total number of classes and the classifier head, respectively. Assuming that $s < K$ is the number of clients participating in each round and $T$ is the total number of communication rounds, the communication cost for federated *full-finetuning* is $2 \cdot (E + d \cdot C) \cdot T \cdot s \cdot 32$ and the cost for federated *linear probing* will be $2 \cdot d \cdot C \cdot T \cdot s \cdot 32$.

For *federated prompt-tuning* methods [54], considering the classifier weights $H = dC$, $T$ rounds of communication, and $s$ clients per round, the communication costs are:

- FedAvg-PT and FedProx-PT share the classifier weights $H = d \cdot C$ and the fixed size prompt pool $P$ for each client of size $d$ at each round. Cost = $2 \cdot (d \cdot C + P \cdot d) \cdot T \cdot s \cdot 32$.
- PFPT shares the classifier weights $H = d \cdot C$ and a variable size prompt pool $P_i$ for each client at each round $i$. Considering the sum of all $P_i$ across all $T$ rounds as $\sum_{i=1}^{T} P_i$, the communication cost is $2 \cdot (d \cdot C \cdot T + d \cdot \sum_{i=1}^{T} P_i) \cdot s \cdot 32$.

For the experiments reported in Table 4, we follow the same training settings as [54] and train for 120 rounds with $s = 10$ clients participating per round. We use a fixed prompt pool size $P = 20$ for FedAvg-PT and FedProx-PT and a variable prompt pool size which is optimized over rounds for PFPT. The sums of the variable prompt pool sizes $\sum_{i=1}^{T} P_i$ across all rounds for CIFAR-100, ImageNet-R, CUB200 and Cars are 1777, 5205, 4736 and 4736 respectively.

Using PFPT [54] on CIFAR-100, the prompt pool size starts from 20 and ends at 10 after 120 rounds leading to improved performance and reduced communication compared to FedAvg-PT and

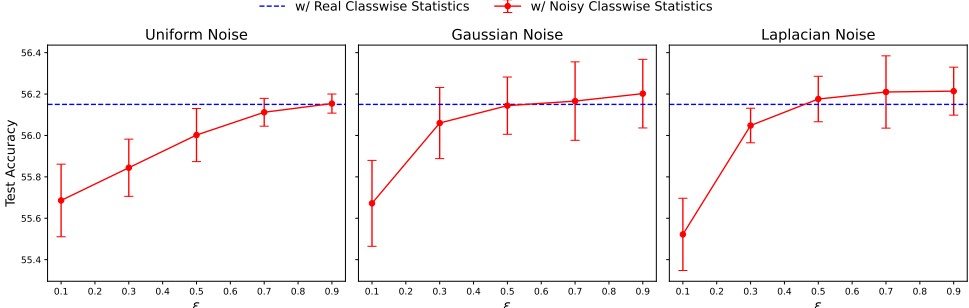

Figure 6: Performance of FedCOF with noisy class statistics on CIFAR-100 using SqueezeNet. The number of clients is fixed at 100 and classes are distributed using a Dirichlet distribution with $\alpha = 0.1$. Results are averaged over five random seeds, each generating different noise in client statistics, and the standard deviation is reported. FedCOF demonstrates robustness to uniform, Gaussian, and Laplace perturbations in class statistics, with performance showing a slight drop as noise, parameterized by $\epsilon$, increases. Lower $\epsilon$ corresponds to higher noise levels in the class statistics.

FedProx-PT. This is consistent with the results from [54]. However, we observe in our experiments on out-of-distribution datasets like ImageNet-R and fine-grained datasets like CUB200 and Cars that the prompt pool size increases over rounds leading to increased communication costs with respect to FedAvg-PT and FedProx-PT. We also see in Table 4 that all existing prompt-tuning methods performs very poorly on fine-grained datasets like CUB200 and Cars.

# I  Improving Privacy Preservation in FedCOF

In this section we discuss additional privacy techniques to prevent the sharing of client class-wise count statistics and class means.

## I.1  Adding Random Noise to Protect Class-wise Statistics.

Our method requires transmitting class-wise statistics to compute the unbiased estimator of the population covariance (Equation (12)) and classifier initialization, similar to other methods in federated learning [29, 35]. In general, transmitting the class-wise statistics may raise privacy concerns, since each client could potentially expose its class distribution. Inspired by differential privacy [10], we propose perturbing the class-wise statistics of each client with different types and intensities of noise, before transmission to the global server. This analysis allows us to evaluate how robust FedCOF is to variations in class-wise statistics and whether noise perturbation mechanisms can effectively hide the true client class statistics. Specifically, we propose perturbing the class-wise statistics as follows:

$$\widetilde{n}_{k,c} = \max(n_{k,c} + \sigma_\epsilon^{\text{noise}}, 0) \tag{23}$$

where $\sigma_\epsilon^{\text{noise}}$ is noise added to the statistics, and $\epsilon$ is a parameter representing the noise intensity. The $\max$ operator clips the class statistics to zero if the added noise results in negative values, which is expected to happen in federated learning with highly heterogeneous client distributions. When clipping is applied, the client does not send the affected class statistic and class mean, and the server excludes them from the computation of the unbiased estimator.

We consider three types of noise:

- *Uniform noise*: $\sigma_\epsilon^{\text{unif}} \sim \mathcal{U}(-(1-\epsilon)n_{k,c}, +(1-\epsilon)n_{k,c})$, proportional to the real class statistics.
- *Gaussian noise*: $\sigma_\epsilon^{\text{gauss}} \sim \mathcal{N}(0, \frac{1}{\epsilon})$, independent of the real class statistics.
- *Laplacian noise* $\sigma_\epsilon^{\text{laplace}} \sim \mathcal{L}(0, \frac{1}{\epsilon})$, which is also independent of the real class statistics.

Lower $\epsilon$ values correspond to higher levels of noise in the statistics.

In Figure 6, we show that the performance of FedCOF is robust with respect to the considered noise perturbation, varying the intensity of $\epsilon \in \{0.1, 0.3, 0.5, 0.7, 0.9\}$. These results suggest that a

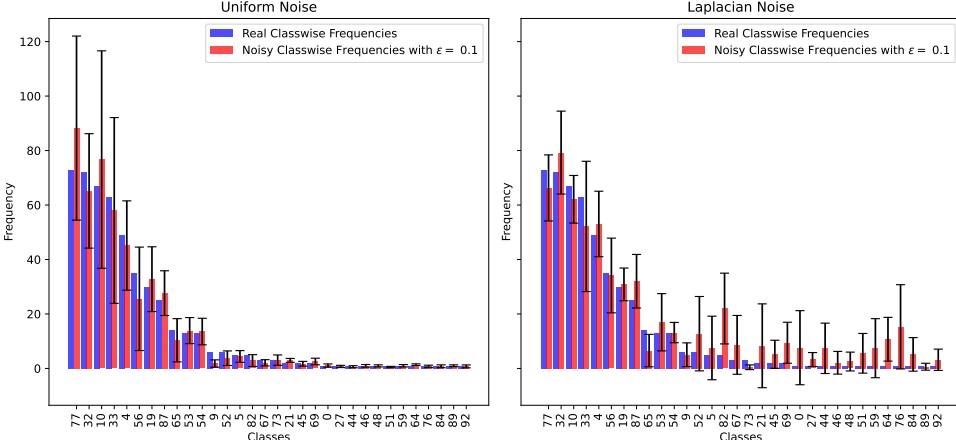

Figure 7: Class frequency distributions for a single client under different noise types: uniform noise (left) and Laplacian noise (right) on CIFAR-100. Both noise types are applied to the real class statistics with the highest noise intensity ($\epsilon = 0.1$). The bar heights represent the average class frequencies, and the error bars indicate the standard deviation across 5 seeds. Real class-wise frequencies and their noisy counterparts are shown for comparison.

differential privacy mechanism can be implemented to mitigate privacy concerns arising from the exposure of client class-wise frequencies. In Figure 7, we provide a qualitative overview of how the proposed Laplacian and uniform noise perturbation affect class-wise distributions.

## I.2  FedCOF with Secure Aggregation Protocols

To incorporate secure aggregation protocols with FedCOF, we propose to use pairwise masking between clients following [3]. Each client $i$ generates a random noise vector $z_{i,j}$ for every other client $j$. Then, each of these clients $i$ computes a perturbation vector $p_{i,j} = z_{i,j} - z_{j,i}$. Similarly, $p_{j,i} = z_{j,i} - z_{i,j}$ resulting in $p_{j,i} = -p_{i,j}$. These perturbation vectors are added to the client statistics before sharing them to the server. When all client statistics are added in the server, the noise cancels out across clients and the server can use the aggregate statistics.

To use such a secure aggregation protocol with FedCOF, the communication cost will increase and become similar to Fed3R. We proceed in two steps:

1. **Secure aggregation of class means and counts.** In this phase, each client sends only class means and counts, adding perturbation vectors to the means and perturbation scalars to the counts to ensure privacy. The server then computes the global class means from the aggregated statistics and sends them back to the clients. Specifically each client $k$ sends:

$$u_{k,c} = q_k + n_{k,c}, \qquad v_{k,c} = p_k + \mu_{k,c} \times n_{k,c},$$

where $n_{k,c}$ and $\mu_{k,c}$ are the original class counts and means; $p_k = \sum_{i \in K, i \neq k} p_{k,i}$ is the total perturbation vector for client $k$, and $\sum_{k=1}^{K} p_k = 0$ (similarly $q_k$ is a perturbation scalar such that $\sum_{k=1}^{K} q_k = 0$).
The server aggregates these quantities, and since the perturbations cancel out, it can compute:

$$\mu_c = \frac{\sum_{k=1}^{K} v_{k,c}}{\sum_{k=1}^{K} u_{k,c}}, \qquad N_c = \sum_{k=1}^{K} u_{k,c},$$

and sends $(\mu_c, N_c)$ back to all clients.

2. **Secure aggregation of class-covariance terms.** Each client computes

$$s_{k,c} = (N_c - 1) n_{k,c} (\mu_{k,c} - \mu_c)(\mu_{k,c} - \mu_c)^\top,$$

and forms

$$S_k = \sum_{c=1}^{C} s_{k,c} + M_k,$$

where $M_k$ is a random matrix such that $\sum_{k=1}^{K} M_k = 0$.

After receiving all $S_k$, the server sums them, and since the noise terms cancel out, it obtains:

$$\sum_{k=1}^{K} S_k = \sum_{k=1}^{K} \sum_{c=1}^{C} (N_c - 1) n_{k,c} (\mu_{k,c} - \mu_c)(\mu_{k,c} - \mu_c)^\top.$$

The final global matrix is then estimated as:

$$\hat{G} = \frac{1}{K-1} \left( \sum_{k=1}^{K} S_k + \sum_{c=1}^{C} K(N_c - 1)\gamma I_d \right) + N\mu_g \mu_g^T$$

$$= \frac{1}{K-1} \left( \sum_{k=1}^{K} \sum_{c=1}^{C} (N_c - 1) n_{k,c} (\mu_{k,c} - \mu_c)(\mu_{k,c} - \mu_c)^\top \right) + \frac{K}{K-1} \sum_{c=1}^{C} (N_c - 1)\gamma I_d + N\mu_g \mu_g^\top$$

$$= \sum_{c=1}^{C} (N_c - 1) \left[ \frac{1}{K-1} \sum_{k=1}^{K} n_{k,c} (\mu_{k,c} - \mu_c)(\mu_{k,c} - \mu_c)^\top + \gamma I_d \right] + N\mu_g \mu_g^\top.$$

where $N = \sum_{c=1}^{C} N_c$ and $\mu_g = \frac{\sum_{c=1}^{C} \mu_c}{N}$. This results in the exact same formulation of the matrix $\hat{G}$ as computed in the proposed FedCOF.

Although this increases training-free communication costs which becomes equivalent to Fed3R, FedCOF offers benefits in terms of accuracy and faster convergence which also reduces overall communication costs when fine-tuning after FedCOF initialization.

## J  Dataset and Implementation Details

**Datasets.** We use the following five datasets in our paper:

- **CIFAR-100** has 100 classes provided in 50k training and 10k testing images.
- **ImageNet-R (IN-R)** is composed of 30k images covering 200 ImageNet classes. ImageNet-R [16] is an out-of-distribution dataset and proposed to evaluate out-of-distribution generalization using ImageNet pre-trained weights. It contains data with multiple styles like cartoon, graffiti and origami which is not seen during pre-training.
- **CUB200** is a fine-grained dataset and has 200 classes of different bird species provided in 5994 training and 5794 testing images.
- **Stanford Cars** has 196 classes of cars with 8144 training images and 8041 test images.
- **iNaturalist-Users-120k** [17] is a real-world, large-scale dataset [48] proposed by [17] for federated learning and contains 120k training images of natural species taken by citizen scientists around the world, belonging to 1203 classes spread across 9275 clients.

In datasets like ImageNet-R and CARS, we also face class-imbalanced situations where there is a significant class-imbalance at the global level.

**Implementation Details.** Here, we provide details on learning rate (lr) used for all fine-tuning experiments with FedAdam. For ImageNet-R and Stanford Cars, we use a lr of 0.0001 for both server and clients for FedNCM, Fed3R and FedCOF initializations. For CUB200, we use a server lr of 0.00001 and client lr of 0.00005 for Fed3R and FedCOF, while for FedNCM, we use a higher lr of 0.0001 for clients. For random classifier initialization with all datasets, we use a higher lr of 0.001 for clients and lr of 0.0001 for server. We use 1 local epoch, for all fine-tuning experiments on 4 datasets. After training-free classifier initialization, we fine-tune the models for 100 rounds. When starting from random classifier initialization, we train more for 200 rounds. When training with FedAvg and random classifier initialization, we use a client lr of 0.005 for all datasets other than inat-120K.

For the linear probing (LP) experiments for the 4 datasets other than iNat-120K, with FedAvg we train for 200 rounds with 1 local epoch and use a client lr of 0.01 and server lr of 1.0 for FedNCM. For Fed3R and FedCOF initializations, we use a client lr of 0.001 and a server lr of 1.0. For LP experiments on iNat-120K, we use 3 local epochs, 30% client participation and train for 5000 rounds. For iNat-120K, we use a client lr of 0.001 for FedAvg-LP without classifier initialization, a client lr of 0.0005 for FedNCM and client lr of 0.00001 for Fed3R and FedCOF.

We use one Nvidia RTX 6000 GPU for all our experiments.

**The FedCOF Oracle (Sharing Full Covariances).** Similar to CCVR [35], we aggregate the class covariances from clients as follows:

$$\hat{\Sigma}_c = \sum_{k=1}^{K} \frac{n_{k,c} - 1}{N_c - 1} \hat{\Sigma}_{k,c} + \sum_{k=1}^{K} \frac{n_{k,c}}{N_c - 1} \hat{\mu}_{k,c} \hat{\mu}_{k,c}^T - \frac{N_c}{N_c - 1} \hat{\mu}_c \hat{\mu}_c^T. \tag{24}$$

For the oracle setting of FedCOF, we use the aggregated class covariance from Equation (24) and apply shrinkage to obtain $\hat{\Sigma}_c + \gamma I_d$ and use it in Equation (8) instead of using our estimated covariances.

# K    Additional Experiments

**Adapting CCVR for Classifier Initialization.**

Table 9: Comparison of FedCOF with CCVR across different datasets and models.

| Dataset | Method | SqueezeNet (d=512) | | MobileNetv2 (d=1280) | | ViT-B/16 (d=768) | |
|---|---|---|---|---|---|---|---|
| | | Acc (↑) | Comm. (↓) | Acc (↑) | Comm. (↓) | Acc (↑) | Comm. (↓) |
| CIFAR-100 | CCVR | **57.5**±0.2 | 3015.3 | 59.6±0.2 | 18823.5 | 72.3±0.2 | 6780.0 |
| | FedCOF (Ours) | 56.1±0.2 | **5.9** | **63.5**±0.1 | **14.8** | **73.2**±0.1 | **8.9** |
| IN-R | CCVR | 36.4±0.2 | 3645.7 | 41.9±0.2 | 22758.8 | 49.3±0.2 | 8197.4 |
| | FedCOF (Ours) | **37.8**±0.4 | **7.1** | **47.4**±0.1 | **17.8** | **51.8**±0.3 | **10.7** |
| CUB200 | CCVR | 51.2±0.1 | 2472.1 | 61.6±0.2 | 15432.7 | 78.7±0.4 | 5558.6 |
| | FedCOF (Ours) | **53.7**±0.3 | **4.8** | **62.5**±0.4 | **12.0** | **79.4**±0.2 | **7.2** |
| Cars | CCVR | 40.9±0.4 | 2767.3 | 36.0±0.4 | 17275.7 | 49.4±0.4 | 6222.5 |
| | FedCOF (Ours) | **44.0**±0.3 | **5.4** | **47.3**±0.5 | **13.5** | **52.5**±0.3 | **8.1** |

Since CCVR [35] was originally proposed to calibrate classifiers after training, we adapt it to our setting and use it as an initialization method. CCVR is similar to the FedCOF Oracle in terms of communication cost as it shares class means and covariances from all clients to the server. CCVR aggregates the class covariances and means to obtain a global class distribution at the server, and then trains a linear classifier on Gaussian features sampled from aggregated class distributions from clients. We show in Table 9 that the proposed FedCOF outperforms CCVR in most settings despite having significantly lower communication cost.

Table 10: Performance comparison of different initialization methods followed by FedAdam optimization for 100 rounds of training.

| Method | ImageNet-R | CUB200 | Cars |
|---|---|---|---|
| FedNCM+FedAdam | 44.7±0.1 | 50.2±0.2 | 48.7±0.2 |
| CCVR+FedAdam | 44.6±0.3 | 51.5±0.2 | 47.9±0.1 |
| Fed3R+FedAdam | 45.9±0.3 | 51.2±0.3 | 47.4±0.4 |
| FedCOF+FedAdam | **46.0**±0.4 | **55.7**±0.4 | **49.6**±0.6 |

We also show in Table 10 that FedCOF initialization is better for further fine-tuning using a pre-trained SqueezeNet that we finetune with FedAdam for 100 rounds after different initialization methods.

**Linear probing after initialization experiments.** We show in Figure 8 that linear probing after FedCOF classifier initialization improves the accuracy significantly compared to FedNCM and is marginally better than Fed3R initialization across three datasets using SqueezeNet.

**Comparison of training-free methods with linear probing.** We also compare with our approach with the training-based federated linear probing without any initialization (where we perform FedAvg and learn only the classifier weights of models) and show in Table 11 that FedCOF is more robust and communication-efficient compared to federated linear probing across several datasets. We follow the same settings as in Table 3. For first 4 datasets, we perform federated linear probing for 200 rounds

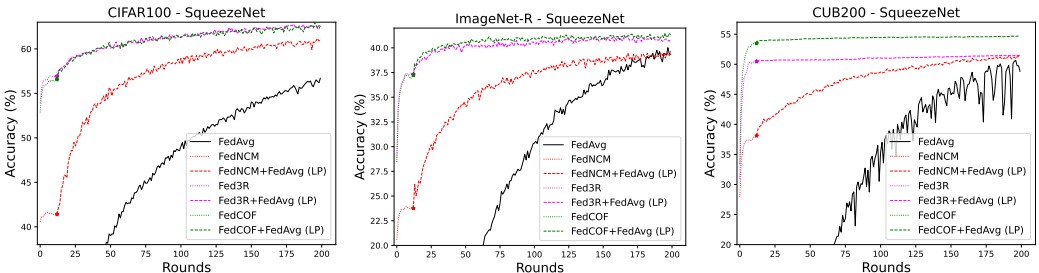

Figure 8: Analysis of the performance with federated linear probing using FedAvg [37].

Table 11: Comparison of different training-free methods using SqueezeNet with training-based Fed-LP (federated linear probing with FedAvg [37] starting with pre-trained model and random classifier initialization) across 5 random seeds. FedNCM, Fed3R and the proposed FedCOF does not involve any training. We show the total communication cost (in MB) from all clients to server. The best results from each section are highlighted in **bold**.

| Method | CIFAR-100 Acc (↑) | CIFAR-100 Comm. (↓) | ImageNet-R Acc (↑) | ImageNet-R Comm. (↓) | CUB200 Acc (↑) | CUB200 Comm. (↓) | CARS Acc (↑) | CARS Comm. (↓) | iNat-120K Acc (↑) | iNat-120K Comm. (↓) |
|---|---|---|---|---|---|---|---|---|---|---|
| Fed-LP | **59.9**±0.2 | 2458 | **37.8**±0.3 | 4916 | 46.8±0.8 | 4916 | 33.1±0.1 | 4817 | 28.0±0.6 | $1.6\times10^6$ |
| FedNCM | 41.5±0.1 | **5.9** | 23.8±0.1 | **7.1** | 37.8±0.3 | **4.8** | 19.8±0.2 | **5.4** | 21.2±0.1 | **111.8** |
| Fed3R | 56.9±0.1 | 110.2 | 37.6±0.2 | 111.9 | 50.4±0.3 | 109.6 | 39.9±0.2 | 110.2 | 32.1±0.1 | 9837.3 |
| FedCOF (Ours) | 56.1±0.2 | **5.9** | **37.8**±0.4 | **7.1** | **53.7**±0.3 | **4.8** | **44.0**±0.3 | **5.4** | **32.5**±0.1 | **111.8** |

with 30 clients per round using FedAvg with a client learning rate of 0.01. For iNat-120k, we train more for 5000 rounds.

**Impact of using pre-trained models.** To quantify impact of using pre-trained models we performed experiments using a randomly initialized model and show in Table 12 that federated training using a pre-trained model significantly outperforms a randomly initialized model using standard methods like FedAvg and FedAdam on CIFAR-10 and CIFAR-100.

Table 12: Impact of using a pre-trained SqueezeNet with different federated learning methods on CIFAR-10 and CIFAR-100. We show the total communication cost (in MB). We train a total of 100 clients with 30 clients per round for 200 rounds in non-iid settings with a Dirichlet distribution of 0.1. When starting from random initialization (no pre-training), we train for 400 rounds.

| Method | Pre-trained | CIFAR-10 Acc (↑) | CIFAR-10 Comm. (↓) | CIFAR-100 Acc (↑) | CIFAR-100 Comm. (↓) |
|---|---|---|---|---|---|
| FedAvg | × | 37.3 | 74840 | 23.9 | 79248 |
| FedAdam | × | 60.5 | 74840 | 44.3 | 79248 |
| FedAvg | ✓ | 84.7 | 37420 | 56.7 | 39624 |
| FedAdam | ✓ | 85.5 | 37420 | 62.5 | 39624 |

**Experiments with ResNet-18.** We perform experiments with pre-trained ResNet-18 in Table 13. For FedAvg and FedAdam, we train for 200 rounds with 30 clients per round. For FedAvg, we train with a client learning rate of 0.001 and server learning rate of 1.0. For FedAdam, we train with a client learning rate of 0.001 and a server learning rate of 0.0001. We show that fine-tuning after FedCOF classifier initialization for 100 rounds outperforms competitive FL methods like FedAdam (which are trained for 200 rounds) by 2.5% on CIFAR-100 and 5.1% on ImageNet-R. The improved performance with FedCOF initialization validates the effectiveness of the proposed method, as it reduces communication and computation costs by half compared to FedAdam and FedAvg and still outperforms them.

Table 13: Comparison of different training-free methods using pre-trained ResNet-18 for 100 clients with training-based federated learning baselines FedAvg [37] and FedAdam [42] starting from a pre-trained model. We train for 200 rounds for FedAvg and FedAdam which uses pre-trained backbone and random classifier initialization. FedNCM, Fed3R and the proposed FedCOF do not involve any training. We also show the performance of fine-tuning with FedAdam after classifier initialization. For fine-tuning experiments we only train for 100 rounds after initialization. We show the total communication cost (in MB) from all clients to server. The best results from each section are highlighted in **bold**.

| | CIFAR-100 | | ImageNet-R | |
| Method | Acc ($\uparrow$) | Comm. ($\downarrow$) | Acc ($\uparrow$) | Comm. ($\downarrow$) |
| --- | --- | --- | --- | --- |
| FedAvg | 67.7 | 538k | 56.0 | 541k |
| FedAdam | 74.4 | 538k | 57.1 | 541k |
| FedNCM | 53.8 | **5.9** | 37.2 | **7.1** |
| Fed3R | 63.5 | 110.2 | 45.9 | 111.9 |
| FedCOF (Ours) | 63.3 | **5.9** | 46.4 | **7.1** |
| FedNCM+FedAdam | 75.7 | 269k | 60.3 | 271k |
| Fed3R+FedAdam | 76.8 | 269k | 60.6 | 271k |
| FedCOF+FedAdam | **76.9** | 269k | **62.2** | 271k |

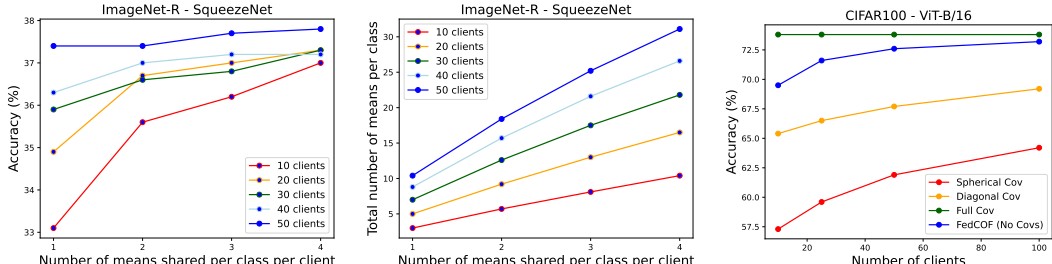

Figure 9: (left) Analysis of FedCOF performance with multiple class means per client on ImageNet-R. (center) Total number of means per class on average that are used to estimate the covariance for FedCOF in Figure 9 (left). (right) Performance comparison of FedCOF with full, diagonal, and spherical covariance matrix communication.

## L Additional Ablations

**Sampling multiple class means.** We perform multiple class means sampling per client using ImageNet-R and show in Figure 9 (left) that using FedCOF with more class means shared from each client improves the performance. We also show in Figure 9 (center) the total number of means used per class on an average in Figure 9 (left) to perform the covariance estimation. The number of means used to estimate each class covariance is less than the total number of clients due to the class-imbalanced or dirichlet distribution used to sample data for clients. This is due to the fact that not all classes are present in all clients.

In most settings we observe that the largest performance improvement occurs when increasing the number of class means per client in scenarios with fewer clients. For instance, with only 10 clients, accuracy improves from 33.1% to 37.0% when increasing from 1 to 4 means per client. In contrast, when 50 clients are available, the improvement is marginal (from 37.4% to 37.8%). This trend is consistent with the theoretical insights in Appendix E: when fewer clients are available, sampling more means per client increases the number of independent statistics, thereby reducing variance and improving the quality of the covariance estimate.

**Communicating diagonal or spherical covariances.** While communicating diagonal or spherical covariances (mean of the diagonal covariance) from clients to server and then estimating the global class covariance from them can significantly reduce the communication cost, such estimates of global class covariance is poor compared to FedCOF. We show in Figure 9 (right) that FedCOF outperforms these covariance sharing baselines when communicating spherical or diagonal covariances.

**Impact of Shrinkage.** We analyze the impact of shrinkage on the estimated class covariances in FedCOF when using a pre-trained SqueezeNet in Table 14. We use a shrinkage $\gamma = 1$ for our experiments with SqueezeNet and ViT-B/16. We observe that shrinkage yields marginal improvement for ImageNet-R and a bit more significant improvement in accuracy of 2.5% on CUB200. This can be attributed to the few-shot settings where the covariance estimation is not very good due to the lack of data and thus fewer clients having access to each of the classes. The use of shrinkage in FedCOF stabilizes and improves the covariance estimation leading to improved accuracy especially in few-shot settings.

Table 14: Ablation showing the impact of using shrinkage in FedCOF using a pre-trained SqueezeNet.

| Dataset | $\gamma = 0$ | $\gamma = 0.01$ | $\gamma = 0.1$ | $\gamma = 1$ | $\gamma = 10$ |
|---|---|---|---|---|---|
| ImageNet-R | 36.53 | 36.98 | 36.96 | 37.25 | 36.07 |
| CUB200 | 51.08 | 51.07 | 51.81 | 53.57 | 53.50 |

**Severe imbalance settings.** We further evaluate FedCOF in settings with more severe class imbalance or heterogeneity across clients (Dirichlet distribution with $\alpha = 0.05, 0.01$) in Table 15 and show that the performance of FedCOF drops a bit with very severe heteregeneity as expected.

Table 15: Analysis of FedCOF performance across settings with varying heterogeneity.

| Class means shared per client | ImageNet-R (100 clients) | | | | Cars (100 clients) | | | |
|---|---|---|---|---|---|---|---|---|
| | $\alpha = 0.5$ | $\alpha = 0.1$ | $\alpha = 0.05$ | $\alpha = 0.01$ | $\alpha = 0.5$ | $\alpha = 0.1$ | $\alpha = 0.05$ | $\alpha = 0.01$ |
| 1 | 38.4 | 37.3 | 36.4 | 33.5 | 45.0 | 44.5 | 43.1 | 39.9 |
| 2 | 38.2 | 37.8 | 37.4 | 35.9 | 44.9 | 44.5 | 43.8 | 42.5 |
| 3 | 38.1 | 37.5 | 37.5 | 36.7 | 44.9 | 44.7 | 44.2 | 43.2 |
| 4 | 38.3 | 37.7 | 38.2 | 37.4 | 45.1 | 44.9 | 44.4 | 43.6 |

We show the impact of class imbalance in different settings in Table 16. Using the most extreme setting ($\alpha = 0.01$), each class is present on 4 clients on an average. Although the extreme settings are less unrealistic, we analyze and evaluate FedCOF in those settings. For instance, the drop in performance of FedCOF on Cars from 44.5 to 39.9 is due to the fact that the global covariance is estimated from around 3.6 sample means instead of around 13.4 sample means. This scenario faces the same issue as fewer clients due to the severe class imbalance. Here, we show that sampling multiple class means per client improves the performance in the extreme settings mitigating the bias in these settings.

Table 16: Total means shared per class on average (the number of clients each class is present in, on average).

| Dataset | $\alpha = 0.5$ | $\alpha = 0.1$ | $\alpha = 0.05$ | $\alpha = 0.01$ |
|---|---|---|---|---|
| ImageNet-R | 36.1 | 17.4 | 11.7 | 4.0 |
| Cars | 24.5 | 13.4 | 9.6 | 3.6 |

## M  Impact Statement

In this paper we propose a highly communication-efficient method for federated learning which exploits pre-trained feature extractors. Reducing communication between clients and the central server is a critical aspect of federated learning to enhance its application in practical scenarios. The proposed method is training-free and thus does not require extensive training or incur excessive computational costs across all client devices like training-based federated learning methods. Our method drastically reduces communication while achieving similar or even better accuracy compared to existing approaches. The proposed initialization can be used with different federated fine-tuning approaches. We believe that our work will advance federated learning applications and make them more efficient.

