# OpenReview forum: "Covariances for Free: Exploiting Mean Distributions for Training-free Federated Learning"
_NeurIPS.cc/2025/Conference — NeurIPS 2025 poster_

### Official Review · Reviewer_Md8N · 2025-06-06

**Clarity:** 4
**Significance:** 2
**Originality:** 1
**Rating:** 4
**Confidence:** 3

**Summary:**

The paper proposes FedCOF, for training-free FL. The goal is, given a pre-trained feature extractor, to set the parameters of the classifier without resorting to standard multi-round gradient based training. Previous methods have done this using feature means or both feature means and outer-products. FedCOF uses both means and outer-products to initialize via a variant of the ridge-regression method proposed by Fed3R but does this without transmitting the outer-product matrices. Instead it estimates the covariance using the standard sample covariance matrix estimator which is unbiased under the assumption that the means are unbiased. It then uses the covariance matrix to recover the outer product matrix, in the process dropping the between class covariants, and computes the ridge regression solution. The paper provides experiments evaluating the performance of the training-free initialization as well as additional experiments examining further finetuning + ablation studies.

**Questions:**

Am I correct in saying that the server sees each individual client statistic $\hat{\mu_{k,c}}$ and $n_{k, c}$?

**Ethical Concerns:**

["NO or VERY MINOR ethics concerns only"]

**Final Justification:**

The authors addressed my main concerns. In particular the argument about fast convergence when fine tuning was valid to me. My main concern was the privacy issue of acting on unaggregated client information (when there is a method that is similar and does not do this) but I think it is still a useful contribution to have the proposed method available as it sits on a different part of the communication vs privacy trade off curve. I find the promised addition of the variant of Fed3R that incorporates the papers proposal of removing $G_{btw}$ a useful one that would really strengthen the paper. This method would then have the added accuracy benefits of FedCOF with better privacy guarantees, but at the cost of higher communication. This would provide practitioners with a useful range of methods.

**Limitations:**

The authors include a valid discussion of the limitations. For me, however, the biggest limitation, namely the privacy issue of working with unaggregated client statistics, is not adequately addressed.

**Quality:**

3

**Strengths And Weaknesses:**

**Strengths**

- The paper is well written. It is clear and easy to follow.

- The methodological contribution is that the client to server communication cost is reduced compared to Fed3R. Additionally, the removal of the between class covariance appears to lead to an increase in empirical performance.

- Theoretical results appear to be sound and the proofs appear to be correct and well written. I would not say the theoretical results are a major contribution, but nor are they claimed to be.

- Experiments are thorough and well conducted. There are a good range of datasets, and test over 3 different realistic models. They additionally conduct experiments showing performance increases when combined with finetuning and linear probing.

**Weaknesses**

My main concern with the paper is the similarity of the method to Fed3R and that the changes made actually make the method less suited to an FL setting in my opinion. I see three main areas in which FedCOF differs from Fed3R, namely Communication Cost, Privacy and Accuracy. FedCOF makes a tradeoff between privacy and communication that I find unfavourable.

Communication cost - loosely speaking FedCOF saves a factor of $d$ in the client to server communication compared with Fed3R. However, in this context I do not find this so critical. Firstly, the actual raw per client communication costs are not that high. For Fed3R in table 3 they are approx 1.1MB, 6.7MB and 2.4MB for the 3 different models. For reference the model sizes are approx 4.8MB, 14MB and 330MB, so significantly larger. Of course it does not hurt to reduce the communication cost in FedCOF, however, for these sizes of uploads it is unlikely to be the bottleneck in the first place, rather sampling devices, scheduling on device, waiting for stragglers etc, is far more impactful. Secondly, we are talking about a single round procedure, so optimizing communication is automatically less impactful than if we were running 1000s of rounds of training. Related to this, the paper shows that multi-round full finetuning of the FedCOF initialization leads to better performance. As a user of FedCOF then of course I would do this, and in this case my overall communication cost is anyway dominated by the model transmission in the fine-tuning phase.

Privacy - As far as I understand, FedCOF requires the server to access each client statistic $\hat{\mu_{k,c}}$ and $n_{k, c}$. This is a very significant privacy issue in my opinion and deviates from the normal FL paradigm of working only with across client aggregate statistics. It makes the method incompatible with client-level differential privacy and we cannot use secure aggregation. Essentially, all formal guarantees of privacy are off. In contrast Fed3R only uses aggregates of client statistics on the server and as such is fully compatible with all privacy addons that are crucial in FL. I see there is a brief discussion of this in Appendix P but I do not find this formal enough. There could be a possibility that FedCOF might work in the setting of data-point level DP in a cross-silo setting, but I do not believe client-level DP, as is the standard for cross-device FL, is possible.

Accuracy - The results tables in the paper show that for some settings FedCOF outperforms Fed3R. It appears to me that Table 2 is the explanation for this, i.e. FedCOF benefits by dropping $G_{btw}$. While this is a nice insight, such a modification should also be simple to make for Fed3R as well, while still maintaining privacy by aggregating client statistics on the server.

**Summary**

I find the paper to be well written, with soundly presented (albeit rather minor) theoretical results and a thorough experimental evaluation. My main concern is that the novel contribution is minor and there exists prior work (Fed3R) that is very similar and that has better privacy characteristics in exchange for a slight increase in communication cost, which I would argue is not so important. FedCOF exhibits better accuracy performance in some, but not all settings, but the improvement that leads to this is independent of the main claimed contribution (sharing only the means) and could be easily implemented into Fed3R.

---

> ### Author Rebuttal · Authors · 2025-07-30
>
> We thank the reviewer for appreciating our paper as well-written with soundly presented theoretical results and a thorough experimental evaluation. We address the reviewer concerns below.
>
> > FedCOF makes a tradeoff between privacy and communication that I find unfavourable.
>
> We appreciate this observation and agree that FedCOF and Fed3R lie at different points on the privacy-communication trade-off curve. Our work explicitly focuses on reducing communication overhead, resulting in a more communication-efficient method that nonetheless achieves competitive—and often superior—performance across a range of experiments. Importantly, federated learning is not one-size-fits-all; practitioners should be able to select algorithms aligned with their specific system constraints and privacy requirements. We believe FedCOF adds valuable diversity to the suite of available methods by offering an effective solution for scenarios in which communication efficiency is critical.
>
> > Communication cost - loosely speaking FedCOF saves a factor of d in the client to server communication compared with Fed3R. However, in this context I do not find this so critical.
>
> Note that the focus of our paper is on *training-free* federated learning using pre-trained models and that our goal is to explore how to do federated learning without sharing client models. Our work aims to achieve similar or better performance with reduced communication without any training or sharing of model parameters. In this context, for the real-world iNat-120K using MobileNetV2, Fed3R has a per-client communication cost of 6.6 MB, while FedCOF reduces that to 0.03 MB. In the training-free setting, where model size does not play a role in communication costs, our proposed method is much more efficient and performs better.
>
> For fine-tuning experiments, the reviewer is correct to point out that communication cost of initialization is relatively small compared to the cost of sharing models during fine-tuning. However, we show in Fig. 3 that FedCOF provides a better starting accuracy (better than Fed3R in most cases) which then significantly reduces the overall number of training rounds required for federated fine-tuning and linear probing. More importantly, our initialization method leads to higher final accuracies on several datasets after fine-tuning (see Fig. 3 where we achieve much faster convergence compared to Fed3R, FedNCM and FedAdam and achieve higher accuracy in far fewer training rounds, thereby saving significant communication and computation costs). For example, on the Cars dataset, FedCOF+FedAdam achieves around 50% accuracy after 50 rounds of training, while FedAdam does not reach this performance after 200 rounds and even Fed3R+FedAdam achieves only around 47% at convergence.
>
> As a final point, we would like to point to the recent trend of parameter-efficient federated fine-tuning [a2,a3] where the per-round communication costs are significantly reduced. For example, recent work [a2] shows that using pre-trained language models (RoBERTa-base), performance similar to full fine-tuning can be achieved using rank-1 LoRA updates and thereby reducing communication cost by 99.8%. In this case, initialization communication costs do contribute to the overall communication costs.
>
> [a2] Towards Robust and Efficient Federated Low-Rank Adaptation with Heterogeneous Clients. ACL, 2025.
>
> [a3] Probabilistic federated prompt-tuning with non-iid and imbalanced data. NeurIPS, 2024.
>
> > Q1. Am I correct in saying that the server sees each individual client statistic $\mu_{k,c}$ and $n_{k,c}$?
>
> Yes, The server receives individual client statistics $\mu_{k,c}$ and $n_{k,c}$.
>
> > Privacy - FedCOF requires the server to access each client statistic $\mu_{k,c}$ and $n_{k,c}$. This is a very significant privacy issue in my opinion and deviates from the normal FL paradigm of working only with across client aggregate statistics.
>
> We understand the reviewer’s concern regarding the privacy implications of our method. However, we clarify that our paper does not focus on the privacy aspects of FL and claims to be more privacy preserving than Fed3R only in scenarios in which secure aggregation protocols are not implemented (lines 133-136) and client feature distributions are exposed to the server. While privacy is an important aspect, our work is not centered on enhancing privacy of existing methods. We will improve the discussion of the privacy aspects of our method (including the limitations with respect to secure aggregation) in any final version of the paper.
>
> We discuss in Appendix P and empirically show how FedCOF performs when client class counts are perturbed inspired by differential privacy. We believe $(\epsilon, \delta)$-differential privacy approaches could be incorporated with our method for more formal privacy guarantees, which is beyond the scope of our paper and we leave for future work.
>
> For incorporating secure aggregation protocols with FedCOF, we propose to use pairwise masking between clients following [a4]. Each client $i$ generates a random noise vector $z_{i,j}$ for every other client $j$. Then, each of these clients $i$ computes a perturbation vector $p_{i,j} = z_{i,j} - z_{j,i}$. Similarly, $p_{j,i} = z_{j,i} - z_{i,j}$ resulting in $p_{j,i} = -p_{i,j}$. These perturbation vectors are added to the client statistics before sharing them with the server. When all client statistics are summed at the server, the noise cancels out across clients and the server can use the aggregate statistics.
>
> To use such a secure aggregation protocol with FedCOF, the communication cost will increase and become similar to Fed3R. We proceed in two steps:
> 1. Sharing only class means and counts by adding perturbation vectors to compute the global class mean and send them back to clients:
>
> + Each client sends the class counts as $u_{k,c} = q_k + n_{k,c}$ and class means as $v_{k,c} = p_k + \mu_{k,c} \times n_{k,c}$, where $p_k = \sum_{i\in K,i \neq k}p_{k,i}$ and $p_{k,i}$ is the perturbation vector for client $i$ and $\sum_k p_k = 0$. $q_k$ is perturbation number similar to $p_k$.
>
> + The server aggregates the $u_{k,c}$ and $v_{k,c}$ to cancel noise vectors cancel out. The global class mean $\mu_c = \frac{\sum_k v_{k,c}}{\sum_k u_{k,c}}$ is computed using these statistics. The server sends $\mu_{c}$ and $N_c = \sum_k u_{k,c}$ back to each client.
>
> 2. Each client computes $s_c =(N_c-1) n_{k,c} (\mu_{k,c}-\mu_c)(\mu_{k, c} - \mu_c)^{\top}$ and then $S_k = \sum_c s_c+M_k$, where $M_k$ is a random noise matrix similar to the perturbation vector $p_k$ so that $\sum_{k}M_{k}=0$. The server receives $S_k$ from all clients and adds them thus canceling out the noise to obtain $\sum_k S_k = \sum_k \sum_c (N_c-1) n_{k,c} (\mu_{k,c} - \mu_c)(\mu_{k, c} - \mu_c)^{\top}$. Now these aggregates could be used to estimate the global matrix as $G = (\sum_k S_k + K(N_c-1)\gamma I_d)/(K-1)+N \mu_g \mu_g^T$ where $N = \sum_c N_c$ and $\mu_g = \frac{\sum_c \mu_c}{N}$. We obtain the exact same formulation of the matrix $G$ as computed in the proposed FedCOF.
>
> Although this increases training-free communication costs (equivalent to Fed3R), we reiterate the benefits offered by FedCOF in terms of accuracy and faster convergence which reduces overall communication costs when fine-tuning after FedCOF initialization.
>
> [a4] Practical secure aggregation for federated learning on user-held data. arXiv:1611.04482 (2016).
>
> > Accuracy -  FedCOF benefits by dropping $G_{btw}$. While this is a nice insight, such a modification should also be simple to make for Fed3R as well, while still maintaining privacy by aggregating client statistics on the server.
>
> We emphasize that removing the between-class scatter $G_{btw}$ is not possible in Fed3R due to the statistics aggregation method used which computes the second order statistics from features of all classes in a client.
>
> Fed3R shares the matrix $G = (N-1) \hat{S} + N \hat{\mu}_g \hat{\mu}_g^\top$ (Eq. 18) where the global covariance matrix $\hat{S}$ can be written as the sum of within-class and between-class covariances. Fed3R could be modified in two ways to incorporate our proposal from Proposition 2:
> 1. To compute $G$ directly using only the within-class scatter, Fed3R has to share the $FF^T$ matrix for each class present in a client. That would significantly change the Fed3R method by increasing its communication budget to $(d+d^2)CK$ which is the same as the FedCOF Oracle and CCVR. For instance, using MobileNetV2 for iNat120K, per client communication of Fed3R increases from 6.6 MB to 38.6 MB, which is almost 3x the size of the model (14 MB).
> 2. An alternative would be sharing the class counts per client in Fed3R (similar to FedCOF) which could be used to compute the class means $\hat{\mu_c}$ and $\hat{\mu_g}$ to compute between-class scatter and then remove it. This change in Fed3R results in a method with the same privacy properties as FedCOF but with higher communication costs.
>
> We highlight that both modifications are possible only due to our derivation in Proposition 2 (see Appendix D), since in standard ridge regression (following the formulation in Eq. 3) the between-class and within-class scatter matrices are not explicit. Thus the accuracy of Fed3R could be improved only based on one of the contributions from our paper.
>
> Even though not the main contribution of this paper, we theoretically derive the ridge regression solution in the form of covariances to enable removal of the between-class scatter. We demonstrate how this change improves performance over standard ridge regression heads across several datasets in FL settings and also in centralized setting (see Tables 7, 8 in Appendix G). This improvement is also evident after fine-tuning for 100 rounds (see Table 5). This improvement is non-trivial and is a novel contribution on using second-order statistics for classification in FL and could definitely be adapted and incorporated in other FL works.

---

> > ### Comment · Reviewer_Md8N · 2025-08-01
> > **Response to Rebuttal**
> >
> > Thanks for the detailed rebuttal.
> >
> > One quick question first: am I missing something or is there perhaps a $\lambda I_d$ missing inside the inverse of eq (8) and Alg 1? Indeed $\lambda$ is an input to the Algorithm but I can't see where it is used.
> >
> > Regarding privacy: I think the point is fair that it can be useful to have a spectrum of methods, and indeed there is a great deal of literature in FL that doesn't have a strict privacy focus (and doesn't adhere to a pattern of working with only aggregates), so while personally I find it more important than savings in communication, the contribution is likely still of interest to the community as a whole.
> >
> > Regarding the discussion on secure aggregation: without getting too much into the details of pairwise masking, the high level idea is essentially that the only reason you can't compute the $\hat{\Sigma_c}$ estimate as an aggregate over the clients initially is that we don't yet know each $\mu_c$. So spend one round to compute $\mu_c$ (which can be computed as an aggregate across clients), then the next round compute $\hat{\Sigma_c}$ as in eq (6) as an aggregate over the clients. To me this feels closer to the variant of Fed3R discussed previously that leaves out $G_{btw}$ than FedCOF since it kills the central idea of only having to share the means.
> >
> > Regarding the variant of Fed3R: As you mention in Option 2, I think in Fed3R one just additionally needs to compute the $N_c$ which is an aggregate quantity across clients, and from that we can compute $G_{btw}$ on the server and subtract it from FF^T as you say. Could you clarify what you mean by saying this method has the same privacy guarantees as FedCOF? In this case all quantities are computed as aggregates over clients I think (at the cost of higher communication of course).
> >
> > Regarding communication: I find the argument that a better initialization might lead to quicker convergence a reasonable one that I didn't initially consider in my review.

---

> > > ### Author Response · Authors · 2025-08-02
> > >
> > > We thank the reviewer for the detailed discussion.
> > >
> > > > One quick question first: am I missing something or is there perhaps a $\lambda I_d$ missing inside the inverse of eq (8) and Alg 1? Indeed $\lambda$ is an input to the Algorithm but I can't see where it is used.
> > >
> > > Eq. (8) is correct. There is no need to introduce an additional hyperparameter $\lambda$ to compute the inverse of $\hat{G}$. In FedCOF, we use the shrinkage hyperparameter $\gamma$ to stabilize the covariance estimation, which already ensures that the matrix $\hat{G}$ is invertible. The mention of $\lambda$ as an input to the Algorithm 1, on the other hand, is a mistake. We thank the reviewer for pointing this out and will remove it the final version of the paper.
> > >
> > >
> > > > Regarding privacy: I think the point is fair that it can be useful to have a spectrum of methods, and indeed there is a great deal of literature in FL that doesn't have a strict privacy focus (and doesn't adhere to a pattern of working with only aggregates), so while personally I find it more important than savings in communication, the contribution is likely still of interest to the community as a whole.
> > >
> > > We appreciate the reviewer's consideration of our point and acknowledgement that the contribution of FedCOF is likely of interest to the community and valuable for users who prioritize reduced communication costs.
> > >
> > > > Regarding the discussion on secure aggregation: without getting too much into the details of pairwise masking, the high level idea is essentially that the only reason you can't compute the $\hat{\Sigma_c}$ estimate as an aggregate over the clients initially is that we don't yet know each $\mu_c$. So spend one round to compute  (which can be computed as an aggregate across clients), then the next round compute $\hat{\Sigma_c}$ as in eq (6) as an aggregate over the clients. To me this feels closer to the variant of Fed3R discussed previously that leaves out $G_{btw}$ than FedCOF since it kills the central idea of only having to share the means.
> > >
> > > The reviewer is correct that the secure aggregation method for FedCOF which we proposed in the previous response precludes our original idea of sharing only the means, since this secure aggregation protocol requires that each client send a $d \times d$ matrix computed as the outer product of differences of class means. We propose this secure aggregation protocol to show that FedCOF can still work with aggregated statistics -- thus improving privacy -- but with the limitation that it changes the original motivation of sharing only class means. We will add this discussion in the final version of the paper.
> > >
> > > > Regarding the variant of Fed3R: As you mention in Option 2, I think in Fed3R one just additionally needs to compute the $N_c$ which is an aggregate quantity across clients, and from that we can compute $G_{btw}$ on the server and subtract it from FF^T as you say. Could you clarify what you mean by saying this method has the same privacy guarantees as FedCOF? In this case all quantities are computed as aggregates over clients I think (at the cost of higher communication of course).
> > >
> > > We proposed a variant of Fed3R as Option 2 and mentioned that it has same privacy properties as FedCOF since we did not consider that $n_{k,c}$ and $B_k$ (the sum of class features per client, defined in Lines 117-119) in Fed3R could be securely aggregated. Without secure aggregation protocols, the server could use $n_{k,c}$ to compute $\mu_{k,c}$ from the matrix $B_k$, which contains the sum of class-wise features ($\mu_{k,c} = \frac{B_{k,c}}{n_{k,c}}$, where $B_{k,c}$ is the $c-$th column of matrix $B_k$). This way, the server has access to both $\mu_{k,c}$ and $n_{k,c}$ similar to FedCOF.
> > >
> > > However, the reviewer is correct that $N_c$ could be computed from aggregates of $n_{k,c}$ across clients (following secure aggregation protocols) and then used to compute the matrix $G_{btw}$. Then, this term can be subtracted from $FF^T$. This option would maintain the same privacy level of Fed3R but with a higher communication cost. We will add this discussion on modified Fed3R in the final version of the paper.
> > >
> > > > Regarding communication: I find the argument that a better initialization might lead to quicker convergence a reasonable one that I didn't initially consider in my review.
> > >
> > > We thank the reviewer for acknowledging this aspect of our contribution. By providing a better intialization, our method enables quicker convergence and thus saves significant computation and communication costs, while still achieving better final performance.

---

> > > > ### Comment · Reviewer_Md8N · 2025-08-04
> > > >
> > > > Thanks for the reply.
> > > >
> > > > I think including this variant of Fed3R that subtracts $G_{btw}$, along with  corresponding experimental results, would be a valuable addition to the paper.  As discussed earlier, there is privacy-communication trade-off curve, and this method should sit at a different point on the curve while still obtaining the same (or potentially better) accuracy which I believe would be useful to practitioners. While it doesn't fit with the central message of reducing communication overhead, it does illustrate the efficacy of the other main contribution, namely removing $G_{btw}$.
> > > >
> > > > I will update my score to reflect the authors rebuttal and subsequent discussion.

---

> > > > > ### Author Response · Authors · 2025-08-07
> > > > >
> > > > > We thank the reviewer for engaging in the discussions. In the final version of the paper, we will include additional discussion on the privacy aspects of our method, including the variant of FedCOF that is compatible with secure aggregation by sharing the outer products of class means, as well as experimental results for the Fed3R variant suggested by the reviewer.
> > > > >
> > > > > We again thank the reviewer for the constructive feedback and for taking our rebuttal into account in the final recommendation.

---

> > > > > > ### Author Response · Authors · 2025-08-08
> > > > > >
> > > > > > As the rebuttal phase comes to an end, we would like to gently remind the reviewer to update the final recommendation and the score to reflect the rebuttal discussions as the reviewer said they would.

---

### Official Review · Reviewer_DZVi · 2025-07-01

**Clarity:** 4
**Significance:** 3
**Originality:** 3
**Rating:** 4
**Confidence:** 4

**Summary:**

This paper proposes a training-free FL framework called FedCOF. FedCOF seeks to compute a closed form ridge regression global solution
shared across multiple clients, given their siloed data and a pre-trained foundation model. The straight-forward solution to this objective requires sharing second-order statistics that are bandwidth consuming. This paper presents an analysis showing that the closed form solution can be (unbiasedly) estimated using only feature mean from each class of each client. FedCOF is shown to achieve better performance and lower communication cost compared to other training-free and training-based FL solutions.

**Questions:**

- In the comparison with FedADAM & FedAvg, it was mentioned that these methods were initialized with a random classifier. What is the architecture of this classifier? Do they use a ridge regression head as well, or do they get a standard non-linear classifier (e.g., 2/3-layer feedforward net)?

- Are the clients using the same random initialization? Otherwise, could it be that the gap in performance (i.e., between FedAdam and FedCOF+FedAdam) is caused by the misalignment of initializations?

**Ethical Concerns:**

["NO or VERY MINOR ethics concerns only"]

**Final Justification:**

This paper addresses a timely problem in federated learning and shows that a training-free approach can perform competitively with training-based methods. The derivation of the unbiased estimator is novel, and the experiments are well chosen to highlight FedCOF’s strengths. I am satisfied with the authors' discussion of the limitations in the rebuttal. Overall, the paper makes a relevant and useful contribution and I'm leaning towards an acceptance decision.

**Limitations:**

Yes

**Paper Formatting Concerns:**

No formatting concern

**Quality:**

3

**Strengths And Weaknesses:**

Strengths:
- This is a meaningful and timely problem. I'm quite surprised to see that training-free federated learning performs competitively with training-based approaches. I also recognize the significance of the communication bandwidth issue. Given these factors, I believe the paper makes a valuable contribution.
- Derivation of the unbiased estimator is sound and novel.
- Writing is generally clear and understandable.
- The paper showcases FedCOF's strong performance across a well-chosen set of tasks, highlighting its potential.

Weaknesses:
- This method seems to require all clients to adopt a ridge regression prediction head, which is where it derives the closed form solution from. I guess this is the same for many previous training-free FL works such as Fed3R. Perhaps the authors could discuss/show empirically what is the upper limit of their method (e.g., is there a task complexity for which the RR head is no longer good enough?)
- Please see the qn section for specific questions/suggestions regarding the experimental setup.

---

> ### Author Rebuttal · Authors · 2025-07-30
>
> We thank the reviewer for appreciating our work as a valuable contribution solving a meaningful and timely problem, with sound and novel derivation of the unbiased estimator, clear writing and strong performance of FedCOF across a well-chosen set of tasks. We are pleased to see that the reviewer recognizes the significance of the communication bandwidth issues in FL applications where our method significantly reduces communication costs compared to existing methods. We address the reviewer questions below.
>
> > This method seems to require all clients to adopt a ridge regression prediction head, which is where it derives the closed form solution from. I guess this is the same for many previous training-free FL works such as Fed3R. Perhaps the authors could discuss/show empirically what is the upper limit of their method (e.g., is there a task complexity for which the RR head is no longer good enough?)
>
> The extreme upper bound would be sharing the real features from clients to server and training a linear classifier (i.e. linear probing) or using a ridge regression head. However, such an upper bound is unrealistic in FL due to privacy concerns. We show another alternative in the form of the CCVR method (see Appendix H) which trains a global linear classifier by sampling features from global class distributions at the server instead of a ridge regression head. We also show in Appendix H how the FedCOF classifier outperforms the linear probing classifier from CCVR in most settings. To the best of our knowledge, among existing works, using a ridge regression head works the best. We believe future work could explore other classifier approaches. Our principal approach of estimating covariances could potentially be used with future methods that exploit second-order statistics.
>
> > In the comparison with FedADAM & FedAvg, it was mentioned that these methods were initialized with a random classifier. What is the architecture of this classifier? Do they use a ridge regression head as well, or do they get a standard non-linear classifier (e.g., 2/3-layer feedforward net)?
>
> Following FedNCM and Fed3R, the architecture of the global classifier at the server is a linear classifier (i.e. a single linear matrix) for all methods in the comparison. For training-free classifier initialization methods, we set the weights in the linear layer using the aggregated client statistics (see lines 104-106 for FedNCM; lines 118-119 for Fed3R; lines 214-216 for FedCOF).
>
> > Are the clients using the same random initialization? Otherwise, could it be that the gap in performance (i.e., between FedAdam and FedCOF+FedAdam) is caused by the misalignment of initializations?
>
> Yes, all clients use the same random initialization to ensure that there is no misalignment due to difference in initialization.

---

> > ### Comment · Reviewer_DZVi · 2025-08-04
> > **Response to authors' rebuttal**
> >
> > Thanks for responding to my questions.
> >
> > I also notice that the benchmark datasets are somewhat in-distribution wrt the pre-trained model. Do you foresee that the linear head would still be sufficient if we were to apply FedCOF on a more OOD dataset (i.e., image data from another domain such as MRI scans)?

---

> ### Author Response · Authors · 2025-08-05
>
> >Do you foresee that the linear head would still be sufficient if we were to apply FedCOF on a more OOD dataset (i.e., image data from another domain such as MRI scans)?
>
> We thank the reviewer for engaging with us in the discussion.
>
> In the experiments presented in the paper, we evaluated FedCOF across several diverse datasets to assess its broad applicability in realistic settings, including scenarios that are out-of-distribution (OOD) with respect to the pre-training dataset ImageNet-1k. Specifically, we included fine-grained datasets such as CUB (bird species) and Stanford Cars (car types), as well as iNaturalist—a large-scale, real-world dataset containing 120k training images of natural species captured by citizen scientists worldwide—which has also been also used to benchmark OOD performance relative to ImageNet-1k[a]. Additionally, we included ImageNet-Rendition (ImageNet-R)[b], a dataset explicitly designed to test OOD generalization using ImageNet pre-trained weights. It contains data with multiple styles like cartoon, graffiti and origami which is not seen during pre-training.
>
> The reviewers suggestion of evaluating our method on image dataset from another domain is interesting. Now, we evaluate training-free methods on two image classification datasets from other domains - one on medical domain (Brain Tumor MRI images) and another on Remote Sensing domain (Resisc45). The Brain Tumor MRI dataset (publicly available in kaggle) consists of 7023 images of human brain MRI images divided into 4 classes: glioma, meningioma, no tumor and pituitary. For remote sensing image scene classification, we use the Resisc45 dataset[c] consisting of 31,500 images categorized into 45 scene classes, capturing a diverse range of environments from a bird’s-eye view. We use the SqueezeNet network pre-trained on ImageNet-1k and follow a highly heterogeneous setting (Dirichlet distribution $\alpha=0.1$) using 100 clients. We present the results from our experiments below.
>
> **Comparison with training-free methods**
>
> |Domain|Dataset|FedNCM|Fed3R|FedCOF|
> |--|--|--|--|--|
> |Medical|Brain Tumor MRI|71.3|86.8|**87.8**|
> |Remote sensing|Resisc45|66.1|79.8|**81.0**|
>
> The results demonstrate that FedCOF achieves strong training-free accuracy even on image datasets from other domains and outperforms existing training-free methods, Fed3R and FedNCM. FedCOF initialization, as shown in Eq. 8, is used to initialize the weights of a linear head
>
> $W^* = \hat{G}^{-1} B$,
>
> and all the clients in the federation employs this standard linear head for classication.
>
> The global performance can be further improved by fine-tuning the feature extractors $\theta_k$ and the classifiers $W_k^*$ for each client $k$, employing a standard federated learning method such as FedAdam. This is even more relevant on challenging OOD benchmarks as we show below.
>
> **Brain Tumor MRI dataset: Accuracy (Fine-tuning the feature extractors and classifiers with FedAdam)**
>
> |Using pre-trained Model|Classifier Initialization|Acc at 10 rounds|Acc at 25 rounds|Acc at 50 rounds|Acc at 100 rounds|
> |-|-|-|-|-|-|
> |❌|Random|48|47.4|56.1|68.3|
> |✅|Random|78.2|83.1|88.8|89.0|
> |✅|FedCOF|**91.4**|**92.3**|**93.3**|**93.9**|
>
> **Resisc45 dataset: Accuracy (Fine-tuning the feature extractors and classifiers with FedAdam)**
>
> |Using pre-trained Model|Classifier Initialization|Acc at 10 rounds|Acc at 25 rounds|Acc at 50 rounds|Acc at 100 rounds|
> |-|-|-|-|-|-|
> |❌|Random|33.6|46.2|52.8|63.6|
> |✅|Random|43.1|66.7|78.8|85.6|
> |✅|FedCOF|**84.2**|**86.6**|**88.1**|**89.3**|
>
> In the results, we observe that pre-trained network is beneficial in general for achieving better performance: from 68.3 to 89.0 on Brain Tumor MRI dataset and from 63.6 to 85.6 on Resisc45 dataset. Finally, we show that FedCOF initialization (Eq. 8) before FedAdam, improves significantly the performance with respect to FedAdam with random classifier initialization on both the benchmarks (from 89.0 to 93.9 on BrainTumor and from 85.6 to 89.3 on Resisc 45) achieving better performance in much less training rounds compared to FedAdam with random classifier initialization.
>
> We demonstrate FedCOF with linear head provides a sufficiently strong starting point on image datasets from other domains. However, overall performance can be further improved by applying full fine-tuning (e.g., with FedADAM) after FedCOF initialization, which benefits in better accuracy and faster convergence thanks to the improved starting point. We thank the reviewer for the discussion and we believe that these results further validates the findings from our paper showing robustness to image datasets from other domains as well.
>
> [a] CADRef: Robust Out-of-Distribution Detection via Class-Aware Decoupled Relative Feature Leveraging, CVPR 2025.
>
> [b] The many faces of robustness: A critical analysis of out-of-distribution generalization, CVPR 2020.
>
> [c] Remote sensing image scene classification: Benchmark and state of the art. Proceedings of the IEEE, 2017.

---

> > ### Author Response · Authors · 2025-08-07
> >
> > We thank the reviewer for the constructive feedback.
> >
> > As the rebuttal phase comes to an end, we would appreciate it if the reviewer could let us know whether our response has  addressed the concerns raised, or if there are any remaining points that may require further clarification.

---

> > > ### Comment · Reviewer_DZVi · 2025-08-07
> > > **Thank you for the rebuttal**
> > >
> > > My concerns have been addressed, and I will remain supportive of this paper.

---

### Official Review · Reviewer_Jmdu · 2025-07-02

**Clarity:** 4
**Significance:** 3
**Originality:** 2
**Rating:** 5
**Confidence:** 4

**Summary:**

This paper introduces FedCOF (Federated Learning with COvariances for Free), a training-free method that leverages pre-trained models and an unbiased estimator of class covariance matrices to initialize the global classifier in federated learning (FL) settings. The key contributions are:
- Proposing a method to estimate class covariances using only client means, which reduces communication costs and mitigates privacy concerns.
- Demonstrating state-of-the-art performance on various FL benchmarks with reduced communication overhead compared to methods that share second-order statistics.
- Showing notable improvements over federated prompt-tuning approaches and effectiveness as an initialization for federated optimization methods.

**Questions:**

- The paper mentions that FedCOF’s performance can degrade with fewer clients and suggests sampling multiple means per client as a remedy. Could the authors provide more detailed empirical results or theoretical analysis to support the effectiveness of this strategy?
  - Are there specific guidelines for choosing the number of means to sample per client?
  - How does the shrinkage factor affect the performance in this case?
- How does FedCOF handle situations with severe class imbalance or heterogeneity across clients? Are there approaches to mitigate potential bias introduced in non-iid settings?
- Are there plans to provide an open-source code to facilitate adoption?

**Ethical Concerns:**

["NO or VERY MINOR ethics concerns only"]

**Final Justification:**

I have decided to maintain my current score and remain supportive of the paper.

**Limitations:**

The authors have discussed some limitations of the work in the conclusion. I think there is no potential negative societal impact.

**Quality:**

4

**Strengths And Weaknesses:**

### Strengths:
1. Overall, the paper is well written and provides a solid theoretical foundation with detailed proofs. Experiments are conducted across multiple datasets (CIFAR-100, ImageNet-R, CUB200, Stanford Cars, iNaturalist-Users-120k) to demonstrate the method's effectiveness.
2. FedCOF eliminates the communication cost associated with sharing second-order statistics by requiring only class means from clients, making it suitable for large-scale FL settings.
3. The method achieves competitive or superior performance compared to existing methods that share second-order statistics (Fed3R, FedNCM).
4. FedCOF can serve as an effective initialization for federated fine-tuning or linear probing tasks, leading to better overall performance.

### Weaknesses:
1. While FedCOF performs well with a large number of clients, its performance may degrade in settings with fewer clients. The authors propose sampling multiple means from each client to mitigate this issue, but more empirical validation is needed.
2. Although the method is designed to handle non-iid data distributions, the impact of extreme class imbalance and client heterogeneity is not fully explored.

---

> ### Author Rebuttal · Authors · 2025-07-30
>
> We thank the reviewer for appreciating our paper as well-written, providing a solid theoretical foundation with detailed proofs, eliminating communication costs making it suitable for large-scale FL settings, with experiments across multiple datasets and an effective initialization method. We address the reviewer questions below:
>
> > The paper mentions that FedCOF’s performance can degrade with fewer clients and suggests sampling multiple means per client as a remedy. Could the authors provide more detailed empirical results or theoretical analysis to support the effectiveness of this strategy? How does the shrinkage factor affect the performance in this case?
>
> From a theoretical perspective, although the estimator $\hat{\Sigma}_c$ is unbiased (Eq. 5), this only ensures that its expected value equals the true covariance $\Sigma_c$ -- not that any individual estimate is accurate. Its *variance* still depends on the number of independent means $K$ sampled: if each client contributes one mean, then $K$ is the number of clients in the federation; if multiple means are sampled per client, $K$ corresponds to the total number of means collected across all clients.
>
> Therefore, increasing $K$, for example by sampling multiple means per client, leads to a tighter concentration of $\hat{\Sigma}_c$ around its expectation, reducing the overall mean squared error (MSE) -- the sum of variance and squared bias (which remains zero since the estimator is unbiased) -- between the estimator and the true covariance matrix. Moreover, when the number sampled means is small relative to the feature dimension $d$, the estimate may be ill-conditioned. The shrinkage term $\gamma I_d$ (Eq. 6) improves numerical stability in such settings, trading a small amount of bias for a significant reduction in variance -- even when only a few clients are available.
>
> From an empirical perspective, in the main paper in Fig. 6 for CIFAR-100 and Fig. 8 for ImageNet-R, we show FedCOF performance under high heterogeneity and class imbalance (non-iid Dirichlet distribution with $\alpha=0.1$) when increasing the number of sampled means per client or the number of clients. Below, we report the detailed accuracy results for ImageNet-R:
>
> **ImageNet-R: Accuracy**
>
> |Class means shared per client|10 Clients|20 Clients|30 Clients|40 Clients|50 Clients|
> |--|--|--|--|--|--|
> |1|33.1|34.9|35.9|36.3|37.4|
> |2|35.6|36.7|36.6|37.0|37.4|
> |3|36.2|37.0|36.8|37.2|37.7|
> |4|37.0|37.3|37.3|37.2|37.8|
>
> In most settings, we observe that the largest performance improvement occurs when increasing the number of class means per client in scenarios with fewer clients. For instance, with only 10 clients, accuracy improves from 33.1% to 37.0% when increasing from 1 to 4 means per client. In contrast, when 50 clients are available, the improvement is marginal (from 37.4% to 37.8%). This trend is consistent with the theoretical insights above: when fewer clients are available, sampling more means per client increases the number of independent statistics $K$, thereby reducing variance and improving the quality of the covariance estimate.
>
> ### Impact of shrinkage in fewer client settings
>
> We show in the table below that using shrinkage (Eq. 6) improves the covariance estimates thereby improving the accuracy:
>
> **ImageNet-R: Accuracy**
>
> |Class means shared per client|Shrinkage|10 Clients||20 Clients||30 Clients||40 Clients||50 Clients|
> |--|--|--|--|--|--|--|--|--|--|--|
> |1|❌|27.1||29.7||32.8||34.5||35.5|
> |1|✅|33.1||34.9||35.9||36.3||37.4|
> |2|❌|30.8||34.0||35.3||36.1||36.7|
> |2|✅|35.6||36.7||36.6||37.0||37.4|
> |3|❌|33.0||35.8||36.2||36.8||37.0|
> |3|✅|36.2||37.0||36.8||37.2||37.7|
>
>
> We observe that shrinkage consistently improves performance, especially when the number of sampled means is small. This aligns with the theory: when the number of means $K$ is low relative to the feature dimension ($d = 512$ in SqueezeNet), the covariance estimate becomes ill-conditioned, and shrinkage stabilizes it. However, as more class means are sampled per client or the number of clients increases, the benefit of shrinkage diminishes, since the total number of means approaches the feature dimension $d$, making the estimate more stable.
>
>
> > How does FedCOF handle situations with severe class imbalance or heterogeneity across clients? Are there approaches to mitigate potential bias introduced in non-iid settings?
>
> While we already use high heterogeneity in all the main experiments of our paper (Dirichlet distribution with $\alpha=0.1$, following several previous works), here we further evaluate with even more severe class imbalance, and thus heterogeneity, across clients (Dirichlet distribution with $\alpha=0.05, 0.01$) and show that the performance of FedCOF drops a bit with very severe heteregeneity as expected:
>
> |Dataset (100 clients)|$\alpha=0.5$|$\alpha=0.1$|$\alpha=0.05$|$\alpha=0.01$|
> |-|-|-|-|-|
> |ImageNet-R|38.4|37.3|36.4|33.5|
> |Cars|45.0|44.5|43.1|39.9|
>
> Below we show the severity of class imbalance in different settings. Using the most extreme setting ($\alpha=0.01$), each class is present on 4 clients on average. This is an extreme and unrealistic setting, but still useful to understand the impact of heterogeneity on our approach. The drop in performance of FedCOF on Cars from 44.5 to 39.9 is due to the fact that the global covariance is estimated from around 3.6 class means instead of around 13.4. In the table below we summarize the average number of class means shared for varying $\alpha$:
>
> **Total means shared per class on average:**
> |Dataset (100 clients)|$\alpha=0.5$|$\alpha=0.1$|$\alpha=0.05$|$\alpha=0.01$|
> |-|-|-|-|-|
> |ImageNet-R|36.1|17.4|11.7|4.0|
> |Cars|24.5|13.4|9.6|3.6|
>
> This scenario faces the same issue as fewer clients due to the severe class imbalance. Our strategy for sampling multiple means per client helps mitigate this effect even in this extreme setting:
>
> **ImageNet-R (100 clients): Accuracy**
>
> |Class means shared per client|$\alpha=0.5$|$\alpha=0.1$|$\alpha=0.05$|$\alpha=0.01$|
> |-|-|-|-|-|
> |1|38.4|37.3|36.4|33.5|
> |2|38.2|37.8|37.4|35.9|
> |3|38.1|37.5|37.5|36.7|
> |4|38.3|37.7|38.2|37.4|
>
> **Cars (100 clients): Accuracy**
>
> |Class means shared per client|$\alpha=0.5$|$\alpha=0.1$|$\alpha=0.05$|$\alpha=0.01$|
> |-|-|-|-|-|
> |1|45.0|44.5|43.1|39.9|
> |2|44.9|44.5|43.8|42.5|
> |3|44.9|44.7|44.2|43.2|
> |4|45.1|44.9|44.4|43.6|
>
> > Are there specific guidelines for choosing the number of means to sample per client?
>
> We propose to use a simple multiple mean sampling strategy following sampling without replacement. We consider the number of means $\mathcal{M}$ to sample as a fixed number which is a hyperparameter. For each class, we take disjoint random sets from $n_{k,c}$ samples in a client and compute the mean for these subsets. We take disjoint sets to avoid computing similar sample means. The only condition we enforce is that clients use atleast 2 samples to compute a mean. If a client does not have atleast $2\mathcal{M}$ samples, we send less than $\mathcal{M}$ sample means. For instance, if a client has 3 samples, we compute and share a single mean.
>
> We observe that the proposed sampling approach improves the performance. However, more sophisticated sampling approaches could be used if the user is interested in improving FL settings with very few clients. One approach could be clients sampling means based on the number of samples $n_{k,c}$ instead of using a fixed number of means to sample from every client. If a client has more samples, it could share more sample means and this would overall share more means and improve the covariance estimates. Future works could optimize this multiple mean sampling approach to better suit fewer client FL settings.
>
> > Are there plans to provide an open-source code to facilitate adoption?
>
> Yes, we will provide the open-source code to enable reproduction of all our experiments including experiments on the real-world iNaturalist-Users-120K.

---

> ### Comment · Reviewer_Jmdu · 2025-08-05
>
> Thank you for the thorough experiment and response. I have decided to maintain my current score and remain supportive of the paper. I believe that incorporating this additional result into the paper would further strengthen its contribution and help convey its significance.

---

### Official Review · Reviewer_WU3D · 2025-07-02

**Clarity:** 3
**Significance:** 3
**Originality:** 3
**Rating:** 5
**Confidence:** 2

**Summary:**

The paper addresses training-free federated learning by proposing FedCOF, a method that estimates per-class covariances using the variance of local class means. This approach leverages pre-trained models to extract features and uses these to derive unbiased global covariances, which are then utilized to obtain a ridge regression classifier. This method avoids sending local covariance matrices, thereby reducing communication costs and potential privacy risks. Experimental results demonstrate that the method excels in both performance and convergence efficiency.

**Questions:**

1. I would like to know if pre-trained models might have negative impacts in certain scenarios. If so, could you please analyze these impacts?
2. The paper primarily discusses classification methods. I am interested in understanding whether the proposed method has broader implications and if it is applicable to other tasks, such as large language models (LLMs).

**Ethical Concerns:**

["NO or VERY MINOR ethics concerns only"]

**Final Justification:**

After carefully reading the authors' response, the other reviews, and the ensuing discussion, I have decided to update my recommendation. The authors have provided satisfactory answers, especially in response to my two main questions (Potential Negative Impacts of Pre-trained Models, and Generalizability of the Method). I believe the work is solid and meets the bar for acceptance. Therefore, I have raised my score to 5 (Weak Accept).

**Limitations:**

yes

**Quality:**

4

**Strengths And Weaknesses:**

**Strengths:**
1. The paper is well-structured and clearly articulated, providing a comprehensive exploration of training-free federated learning using feature extractors. It addresses a pertinent and intriguing problem in the field.

2. The proposed estimator for covariances is both innovative and theoretically sound. The authors have conducted extensive experiments to validate the effectiveness of their approach, demonstrating significant improvements over existing training-free methods.

3. The paper thoroughly considers aspects such as communication cost, convergence, and privacy, reflecting meticulous attention to detail. The derivations are presented in a clear and comprehensible manner, enhancing the paper's accessibility.


**Weaknesses:**

1. The method primarily focuses on classification tasks. Expanding the discussion to include its applicability to a wider range of tasks would enhance the paper's impact and demonstrate the broader feasibility of the approach.

---

> ### Author Rebuttal · Authors · 2025-07-30
>
> We thank the reviewer for appreciating our paper as well-structured and clearly articulated, with innovative and theoretically sound estimator for covariances, clear and comprehensive derivations, and extensive experiments. We address the reviewer questions below.
>
> > I would like to know if pre-trained models might have negative impacts in certain scenarios. If so, could you please analyze these impacts?
>
> In our experiments across several datasets and architectures, we have not found any negative effects of using pre-trained models. While the vast majority of works in FL observed several positive effects of using pre-trained models like significantly reducing the impact of data heterogeneity and reducing the communication and computation budget by achieving early convergence, we found one recent work [a1] which shows that using a ResNet18 pre-trained on CIFAR-10 can sometimes have  negative effects in settings with high data heterogeneity. We show in Appendix M that using pre-trained weights is always better across several datasets.
>
> [a1] Kim et al., Fedfn: Feature normalization for alleviating data heterogeneity problem in federated learning. In International Workshop on Federated Learning in the Age of Foundation Models in Conjunction with NeurIPS 2023.
>
> > The paper primarily discusses classification methods. I am interested in understanding whether the proposed method has broader implications and if it is applicable to other tasks, such as large language models (LLMs).
>
> We thank the reviewer for the question. Our method is designed for classification tasks, which is the primary focus of the paper. However, we believe that training-free approaches like FedCOF could be extended to LLMs when a classification head is incorporated—for example, in tasks such as multiple-choice question answering with models like LLaMA.
>
> That said, applying training-free methods to non-classification tasks (e.g., generation or summarization) remains an open challenge. Exploring this broader applicability is a promising direction for future work, especially given the communication efficiency of training-free methods, which could make them appealing in resource-constrained settings.

---

> > ### Comment · Reviewer_WU3D · 2025-08-03
> > **Response to Rebuttal**
> >
> > Thank you to the authors for the detailed response, which has addressed my concerns.

---

### Note · Authors · 2025-08-14

We thank all reviewers for their constructive feedback. We are pleased to see that the reviewers find the paper as well-written (**WU3D, Jmdu, DZVi, Md8N**), with solid theoretical foundations eliminating communication costs making it suitable for large-scale FL settings (**Jmdu**), sound and novel unbiased estimator addressing the communication bandwidth issue (**DZVi**), with sound theoretical results and thorough experimental evaluation (**Md8N**).

In the rebuttal, we addressed the concerns of all reviewers and we are glad to see their positive continued support for our paper.

With reviewer **WU3D**, we discussed the impact of using pre-trained models in FL and broader applications of the proposed method.

For reviewer **Jmdu**, we provided theoretical analysis of the proposed strategy of sampling multiple class means per client with more discussions on empirical results, the impact of shrinkage in fewer-client settings, and more discussions on settings with severe heterogeneity.

For reviewer **DZVi**, we clarified how the classifier initialization is performed in the fine-tuning baselines and provided empirical results showing improved performance of FedCOF in datasets from other image domains which also achieves consistently better performance on further fine-tuning after FedCOF initialization.

Regarding the concerns raised by reviewer **Md8N**, we engaged in an in-depth discussion on the privacy aspects of our method, the communication–privacy trade-off, and the accuracy improvements brought by removing the between-class covariance that we propose in our method  (FedCOF). During the rebuttal, we clarified that FedCOF and the baseline Fed3R lie at different points on the privacy–communication trade-off curve with FedCOF outperforming Fed3R in most cases across several datasets and domains. We also presented an approach for secure statistics aggregation compatible with FedCOF, and discussed another variant of Fed3R (by removing $G_{btw}$ following our proposition) to achieve improved performance. We also provided additional arguments and evidence showing how the better initialization provided by FedCOF leads to faster convergence, reducing communication costs, and improved final performance compared to Fed3R and FedNCM after finetuning. We demonstrated this in the paper and further highlighted it in the rebuttal. We are glad that the reviewer acknowledged these points.

We again thank all reviewers for engaging in the subsequent discussions.

---

### Decision · Program_Chairs · 2025-09-17

**Decision:**

Accept (poster)

**Comment:**

In order to solve the linear regression problem perfectly, typically we need to have access to covariance matrices of the features. This paper proposes a communication-efficient unbiased estimator of the covariance, and leverages the estimated statistics to initialize the linear classifier on top of pre-trained feature extractors in federated learning. Empirical improvements are convincing, and the algorithmic modification is simple but well-justified. The author rebuttal addressed the reviewers’ major concerns, and all reviewers recommend acceptance. I’d encourage the authors to incorporate the feedback such as generalizing to other non-classification tasks and potential privacy risks into the revision of the paper.